# The characteristics of tides and their effects on the general circulation of the Mediterranean Sea

Bethany McDonagh[1,2], Emanuela Clementi[1], Anna Chiara Goglio[1], and Nadia Pinardi[2]

[1]CMCC Foundation - Euro-Mediterranean Center on Climate Change, Italy
[2]Department of Physics and Astronomy, University of Bologna, Italy

**Correspondence:** Bethany McDonagh (bethany.mcdonagh@cmcc.it), Emanuela Clementi (emanuela.clementi@cmcc.it)

**Abstract.** The effects of tides on the Mediterranean Sea's general circulation, with a particular focus on the horizontal and vertical currents, are investigated using twin simulations with and without tides. Amplitudes of tides in the region are typically low, but an analysis of the potential and kinetic energy demonstrates that tides have effects across many spatial and temporal scales in the basin, including nonlinear effects at short periods (less than one day) with high kinetic energy peaks at near-inertial, basin modes and tidal frequencies. Internal tidal waves are also revealed below 100m. Tides are found to amplify several basin modes of the Mediterranean Sea, broaden several tidal frequency energy spectra bands, as well as interact energetically with near-inertial waves. Tides increase the mixed layer depth in the Mediterranean Sea, particularly in the deep and intermediate water formation areas of the Western and Eastern basins. The addition of tides, the cases considered, does also enhance Western Mediterranean Deep Water formation.

## 1 Introduction

Tidal forcing is a rather recent addition to large-scale circulation models of the ocean, since horizontal and vertical resolutions have recently become fine enough to allow for an explicit and more accurate representation of tides. This has given rise to novel opportunities to analyse tides and their impacts on the ocean circulation. For an overall review of the topic, see Arbic (2022). Tides are now considered to be essential components of the large scale circulation (St. Laurent et al., 2002; Müller et al., 2010; Melet et al., 2016); their global average input energy is 3.5 TW (Simmons et al., 2004), the second largest after winds (Ferrari and Wunsch, 2010), and they generate internal tides that then drive internal mixing (de Lavergne et al., 2020). Tides are also an important phenomenon in coupled numerical models, which represents a new opportunity in high-resolution modelling of the ocean and atmosphere (Arbic, 2022), and long-term simulations.

Internal tides are characterised by relatively large vertical velocities with respect to the Ekman suction and pumping, while their horizontal wavelength is similar to the general circulation scales (Oddo et al., 2023). Recently, observational and modelling studies have begun to advance our understanding of internal tides in several regions of the Mediterranean Sea. The presence of internal tides generated in the Gibraltar Strait is discussed in Morozov et al. (2002), Vlasenko et al. (2009), and Hilt et al. (2020). Diurnal internal tide oscillations are generated around the Adriatic Sea islands in observational studies (Mihanović et al., 2009), leading to baroclinic trapped waves. In a recent paper, Oddo et al. (2023) found observational and

modelling evidence of internal tides at the Sicily-Malta escarpment due to the diurnal tidal components, connected to the bathymetric gradients.

For the Mediterranean Sea, several authors have depicted the importance of tidal motion in the Gibraltar Strait (Armi and Farmer, 1985; Candela et al., 1990; Harzallah et al., 2014; Naranjo et al., 2014; Sannino et al., 2015). Armi and Farmer (1985) and Farmer et al. (1988) first observed the hydraulic control points that are induced by tides, and the importance of tidal dynamics and their variability in the region have been more recently discussed by Vázquez et al. (2006), Sánchez-Román et al. (2012), García-Lafuente et al. (2013), and Hilt et al. (2020). Naranjo et al. (2014) and Harzallah et al. (2014) found that tides at the Strait of Gibraltar: (1) intensify the high frequency dynamics in the Gibraltar Strait, and (2) increase the salinity and, to a lesser extent, decrease the temperature of Atlantic inflowing waters through the enhancement of mixing, affecting the water mass formation processes further downstream from the Strait. Tides also change the Mediterranean water outflow, as demonstrated by Izquierdo and Mikolajewicz (2019), where outflowing water is modified with tides, demonstrating the role of tides in the spreading of outflowing water. Moreover, Ambar and Howe (1979) found that tides increase the variability of outflowing salinity.

Other features of the Mediterranean Sea that bear importance when discussing tides are the free barotropic oscillations induced by atmospheric pressure and wind forcing. The Mediterranean Sea's first barotropic basin mode is at 38.5 hours (Schwab and Rao, 1983), beyond that of diurnal tides, but higher modes also exist at 11.4, 8.4, and 7.4 hours (Schwab and Rao, 1983), and 8 hours (Lamy et al., 1981; Lozano and Candela, 1995). Many of these free oscillations could be affected or enhanced by tides, especially considering their proximity to tidal frequencies. The Adriatic Sea is characterised by seiches or free barotropic oscillations close to tidal frequencies: at 10.7 hours and 21.9 hours (Medvedev et al., 2020; Leder and Orlić, 2004; Schwab and Rao, 1983), and at 12.0 hours (Lozano and Candela, 1995). Medvedev et al. (2020) used tide gauge data to explicitly find that tides resonate with the Adriatic Sea modes, while Lozano and Candela (1995) discussed the interaction of the M2 tide with modes in the Gulf of Gabes, Adriatic Sea, Aegean Sea, and the Mediterranean Sea. Palma et al. (2020) found additionally that spectra of kinetic energy in the Sicily Channel are enhanced at 8 hours and 6 hours due to the nonlinear effects of tides. These works demonstrate the potential importance of interactions between tides and higher frequency features of the Mediterranean Sea, but the interaction between tides and barotropic oscillations in the Mediterranean Sea have not been investigated using a state-of-the-art numerical model.

In the North Atlantic, the addition of tides was found to improve the representation of the mixed layer depth and deep water formation in the Labrador Sea (Müller et al., 2010). Lee et al. (2006) found that in the same region, sea surface density was increased by the addition of tides to their model and found that this could enhance ventilation and overturning. Internal wave frequencies around 8 hours were also found by van Haren et al. (2014) in the northwestern Mediterranean Sea. These relationships have, to our knowledge, not been investigated in detail in the Mediterranean Sea.

In this work, we further characterise the tidal large scale circulation effects in the overall Mediterranean Sea. Analysis of the effects of tides on circulation has been carried out in the Western Mediterranean basin by Naranjo et al. (2014), in the Gibraltar Strait by Sannino et al. (2014) and Gonzalez (2023), in the Sicily Strait by Oddo et al. (2023) and Gasparini et al. (2004) and in

the whole Mediterranean basin, the general circulation without tides was analysed by Pinardi et al. (2019). As far as we know, the effects of tides on the circulation of the entire Mediterranean Sea has not been extensively investigated.

To do this, we analyse the differences between a general circulation ocean model with and without tides, focusing on changes in the spectra, in the horizontal and vertical dynamics both across the basin on average and in several water mass formation regions. Furthermore, we show the impact of tides in the Eastern Mediterranean Sea and the amplification of different basin modes by a tidal resonant mechanism.

In the Mediterranean Sea there are four key regions of deep and intermediate water formation (see Figure 1): the Gulf of Lion (Western Mediterranean Deep Water, WMDW) and South Adriatic (Eastern Mediterranean Deep Water, EMDW) for deep water formation, and the Cretan Sea (Cretan Deep Water, CDW, and Cretan Intermediate Water, CIW) and Rhodes Gyre (Levantine Deep Water, LDW, and Levantine Intermediate Water, LIW) for intermediate water formation (Pinardi et al., 2015). In these regions the vertical velocities are enhanced, and this paper will analyse the differences of water mass formation rates with and without tides in detail.

Section 2 describes the model and observational data used and the analysis methods, while Sections 3-7 contain the model results and conclusions as follows:

- Section 3: Sea level energy spectra

- Section 4: Kinetic energy spectra

- Section 5: Mixed layer depth and water mass formation

- Section 6: Temperature and salinity

- Section 7: Conclusions

## 2 Data and methods

The general circulation model used is NEMO v3.6, following the implementation of the Mediterranean Sea forecasting system operational in the framework of the Copernicus Marine Service (Coppini et al., 2023; Clementi et al., 2021). The area covered by the model is shown in Fig. 1. The model without tidal forcing is validated in Coppini et al. (2023) and as part of the model information in Clementi et al. (2019), while the experiment with tides is validated in the Quality Information Document in Clementi et al. (2021).

The model has $\frac{1}{24}^{\circ}$ uniform horizontal resolution with 141 uneven vertical levels, using z* vertical coordinates and partial steps. This model is eddy-resolving, since the Rossby radius of deformation is of the order of 10km in the Mediterranean Sea and the horizontal resolution of the model is approximately 4km. Atmospheric forcing fields are given by the six-hour European Centre for Medium-Range Weather Forecasts (ECMWF) analyses, at a horizontal resolution of $\frac{1}{8}^{\circ}$ (2015-2018) and $\frac{1}{10}^{\circ}$ (2019-2021) and they are used to compute momentum, water and heat fluxes using specifically designed bulk formulae (Pettenuzzo et al., 2010). Lateral open boundary conditions are used in the Atlantic Ocean and Dardanelles Strait, which are

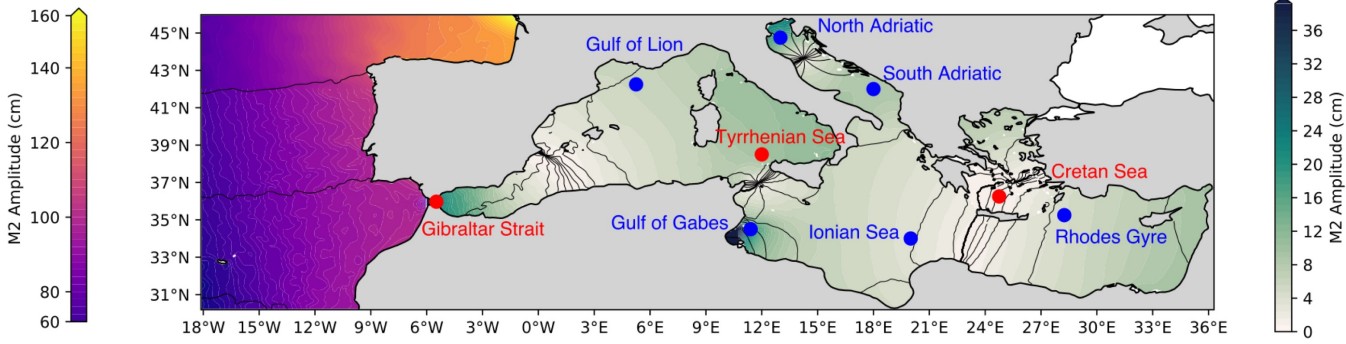

**Figure 1.** Map of the M2 tidal component amplitude in the Atlantic Ocean (contours, colourbar on the left) and Mediterranean Sea (contours, colourbar on the right) and M2 phase in black contour lines, with highlighted points in regions of open ocean dense water formation (Gulf of Lion, South Adriatic, Cretan Sea, and Rhodes Gyre) and regions of high tidal amplitude (Gibraltar Strait, Gulf of Gabes, North Adriatic Sea). Additional points that are studied further in Section 4 are also shown (Tyrrhenian Sea, Ionian Sea).

90 provided by the Copernicus Marine global analysis and forecast system (Galloudec et al., 2022) for the Atlantic Ocean, and a mixture of the aforementioned global model and daily climatology derived from a Marmara Sea box model (Maderich et al., 2015) at the Dardanelles Strait boundary. Further details of the boundary conditions are in Clementi et al. (2021).

 Additionally, monthly mean climatological freshwater inputs from 39 rivers are added to the surface layer. Several datasets are used for this: the Global Runoff Data Centre dataset (Fakete et al., 1999) for the Po, Ebro, Nile, and Rhône rivers, the dataset

95 from Raicich (1996) for the Vjosë and Seman rivers, the UNEP-MAP dataset (Demiraj et al., 1996) for the Buna and Bojana rivers, and the PERSEUS dataset (Deliverable of Perseus, 2012) for the remaining 32 Mediterranean rivers which have a mean run-off larger than 50 $\mathrm{m^3s^{-1}}$. The bathymetry used is the GEBCO 30-sec bathymetry (GEBCO Bathymetric Compilation Group 2014, 2014), interpolated onto the model grid.

 In the chosen configuration, vertical diffusivity depends on the Richardson number, calculated according to Pacanowski and

100 Philander (1981). The Richardson number is a ratio of buoyancy to horizontal shear:

$$Ri = \frac{N^2}{\frac{\partial \bar{v}}{\partial z}^2} \tag{1}$$

where $N^2$ is the Brunt-Väisälä frequency, and $\frac{\partial \bar{v}}{\partial z}^2$ is the vertical shear of horizontal flow. Enhanced vertical viscosity and diffusivity are used where layers are unstable ($N^2 \leq -1 \times 10^{-12}$ $\mathrm{s^{-2}}$) imposing a vertical diffusivity coefficient of $10$ $\mathrm{m^2s^{-1}}$. Vertical viscosity and diffusivity are then calculated according to:

105 $$A^{vT} = \frac{A^{vT}_{ric}}{(1+aRi)^n} + A^{vT}_b \tag{2}$$

| Tidal component | Period (hours) | Description |
|:---:|:---:|:---:|
| M2 | 12.421 | Principal lunar semidiurnal tidal constituent |
| S2 | 12.000 | Principal solar semidiurnal tidal constituent |
| K1 | 23.934 | Lunisolar diurnal tidal constituent |
| O1 | 25.819 | Lunar diurnal tidal constituent |
| N2 | 12.658 | Larger lunar elliptic semidiurnal tidal constituent |
| P1 | 24.066 | Solar diurnal tidal constituent |
| Q1 | 26.868 | Larger lunar elliptic diurnal tidal constituent |
| K2 | 11.967 | Lunisolar semidiurnal tidal constituent |

**Table 1.** Tidal constituent components used in the model for this work, with their respective periods and astronomical descriptions.

$$A^{vm} = \frac{A^{vT}}{(1 + aRi)} + A_b^{vm} \tag{3}$$

where $A^{vT}$ and $A^{vm}$ are the vertical eddy viscosity and diffusivity respectively, and $A_{ric}^{vT}$ (the maximum viscosity value, $100 \times 10^{-4}$ m$^2$s$^{-1}$), $a$, and $n$ are adjustable parameters. In the model runs used, $a$ and $n$ are 5 and 2 respectively. Background vertical eddy viscosity and vertical eddy diffusivity values are $A_b^{vT} = 10^{-7}$ m$^2$s$^{-1}$ and $A_b^{vm} = 1.2 \times 10^{-6}$ m$^2$s$^{-1}$ respectively, which are typical values in the case of constant viscosity and diffusivity, representing diapycnal mixing from the breaking of internal waves. The constant values used here follow the NEMO implementation by Tonani et al. (2008).

Twin experiments are presented in this work with and without the representation of tides. In the experiment including tides, the 8 major tidal components for the Mediterranean Sea (M2, S2, K1, O1, N2, Q1, K2, P1) are represented, which are detailed further in Table 1. The M2 amplitude is displayed in Fig. 1 showing the well-known high amplitude areas at the Gibraltar Strait, the Gulf of Gabes and the Northern Adriatic Sea. Moreover, the Atlantic Ocean lateral open boundary fields include tidal forcing from TPXO9 (Egbert and Erofeeva, 2002).

Other than the addition of the tidal forcing itself, there are some further differences between the tidal and non-tidal modelling set-ups (see Table 2), with the primary difference being the time integration method. Both experiments use a split-explicit free surface formulation proposed in Shchepetkin and McWilliams (2005), solving the free surface equation and the associated barotropic velocity equations with a smaller timestep than the one used for the three-dimensional prognostic variables. The non-tidal experiment uses forward time integration, in which the external mode (barotropic timestep) is integrated between the current and the subsequent baroclinic time steps. This was not stable in the experiment including tides, so this instead uses a centred integration scheme, as recommended by the NEMO manual (Madec et al., 2019), where the baroclinic to barotropic forcing term given at the actual time step becomes centred in the middle of the integration window. The tidal experiment uses a shorter baroclinic timestep, 120s, rather than 240s in the experiment without tides, which was changed for stability purposes. The effects of these changes are negligible compared to the effects of adding tidal forcing to the model (see supplementary materials). The bathymetry was adjusted at several points in the tidal experiment for stability reasons: in the Bay of Biscay,

|  | **Experiment without tides** | **Experiment with tides** |
|---|---|---|
| Baroclinic timestep | 240s | 120s |
| Time integration scheme | Forward | Centred |
| Tides | No tidal forcing | 8 tidal components: M2, S2, K1, O1, N2, P1, Q1, K2 |
| Boundary conditions | No tidal signal at the boundary | Additional tidal signal at the Atlantic boundary: horizontal currents and sea surface height |
| Bathymetry | GEBCO 2014 interpolated onto model grid | Topography changed in 3 grid points at the boundary in the Bay of Biscay and 4 grid points in the Adriatic Sea |

**Table 2.** Summary of the key differences between the tidal experiment and the experiment without tides.

at the northernmost part of the model domain along the French coast, and several points were modified along the Croatian
coastline to avoid generation of spurious tidal signals due to islands and complex bathymetry. All of the above changes in the
tidal experiment compared to the experiment without tides are implemented as in Agresti (2018).

The model was integrated for seven years (2015-2021), starting from climatological temperature and salinity initial conditions derived from the winter climatology (2005-2012) of the World Ocean Atlas (Boyer et al., 2013). The first two years are
removed from the following analysis because they are considered as a spin-up period.

## 3 Sea level energy spectra

Figure 2 shows a spectrum of hourly sea surface height (SSH) in the Mediterranean Sea for five years, for both tidal and non-
tidal experiments. The diurnal and semidiurnal tidal components represent the major differences between the two solutions,
as expected. Further differences between the two models away from these tidal frequencies are also visible. Tides appear to
modulate the spectrum at the mesoscales: energy is reduced in the tidal model at frequencies with a period longer than two
days.

Peaks at both 24 hours and 12 hours are apparent, both with and without tides. In the case without tides, the energy peaks can
be attributed to basin modes excited by atmospheric pressure forcing, as found by Oddo et al. (2014) using a similar NEMO
configuration. In the semidiurnal range, the tidal model introduces a broad peak around 12 hours, as well as peaks at the individual tidal component frequencies. This is due to the amplification by tides of the basin modes near and at these frequencies.
We argue that the broad 12h energy peak in Figure 2 in the tidal run is composed of the amplified 11.4h Mediterranean Sea
basin mode energy (Schwab and Rao, 1983) and the 12h Adriatic/Aegean seas mode (Lozano and Candela, 1995). Moreover,
interactions between internal tides and other mesoscale phenomena such as eddies could be affecting the internal tidal kinetic
energy, as discussed in Guo et al. (2023), but a dedicated analysis would be required to confirm whether this is the case in the

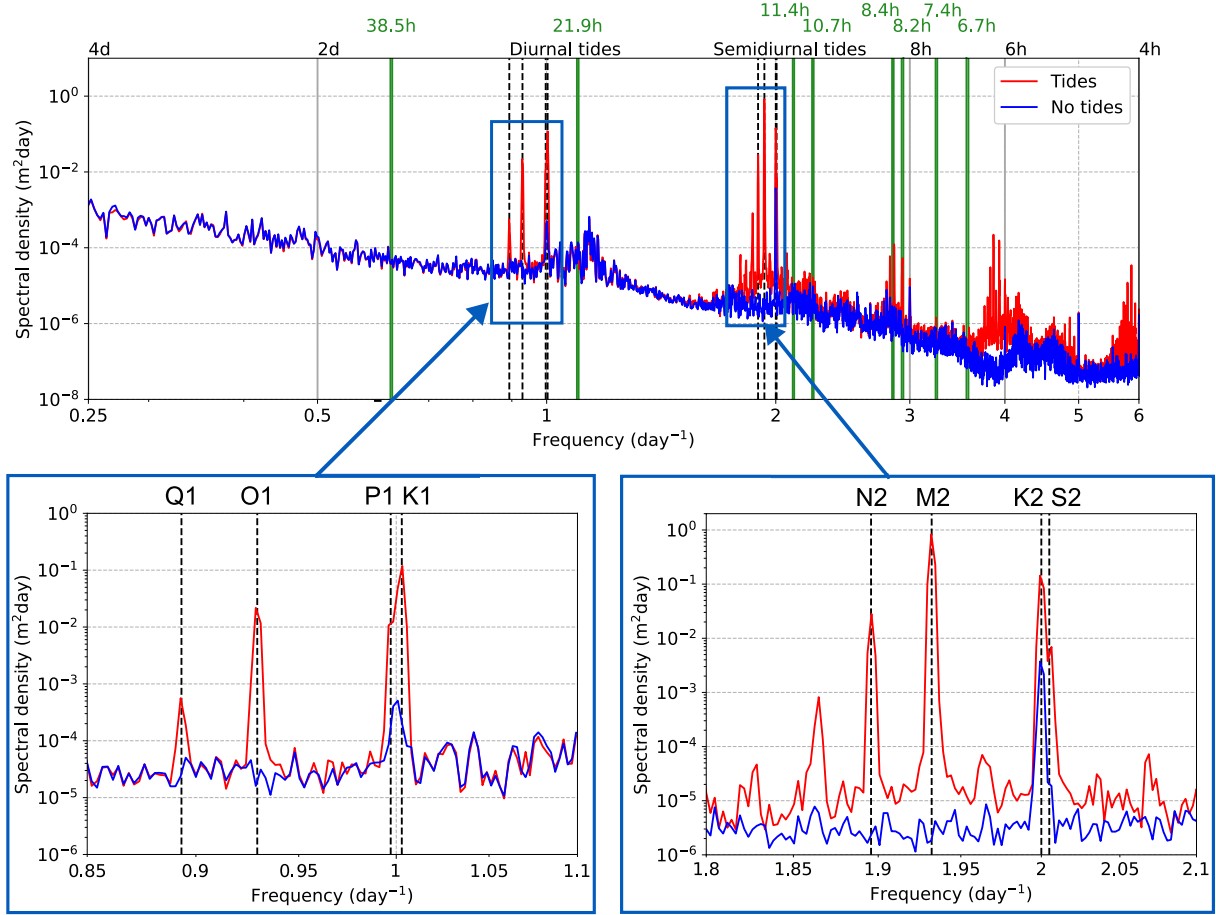

**Figure 2.** Energy density spectrum of hourly mean sea surface height in the Mediterranean Sea for the period 2017-2021, for the experiment without tides (blue) and with tides (red). The basin-mean spectrum is calculated as an area-weighted mean of the periodogram at each gridpoint in the Mediterranean Sea. Boxes below the figure show the key diurnal and semidiurnal ranges to highlight the representation of each individual tidal component. Green lines on the figure indicate frequencies of known barotropic oscillations in the Mediterranean Sea and its sub-basins.

Mediterranean Sea. Furthermore, the spatial and temporal averaging of model outputs has likely blurred these peaks slightly.
Peaks are also seen at 8h and 6h, which correspond to the kinetic energy frequencies in the Sicily Strait, noted by Palma et al. (2020) where these peaks were considered to come from non-linear effects of tides. Further peaks that are enhanced by tides align with the Western Mediterranean basin mode of 8.4h discussed in Schwab and Rao (1983), and the Gulf of Gabes mode at 8.2h (Lozano and Candela, 1995). Schwab and Rao (1983) also noted a fourth Mediterranean Sea mode at 7.4h and the third Adriatic Sea mode at 6.7h, but these are not visible in the basin-mean spectrum of Fig. 2 and are not affected by tides. Overall,
there are many oscillations in the Mediterranean Sea that are interacting with tides leading to their enhancement, although not every barotropic oscillation mentioned in the literature is affected by tides.

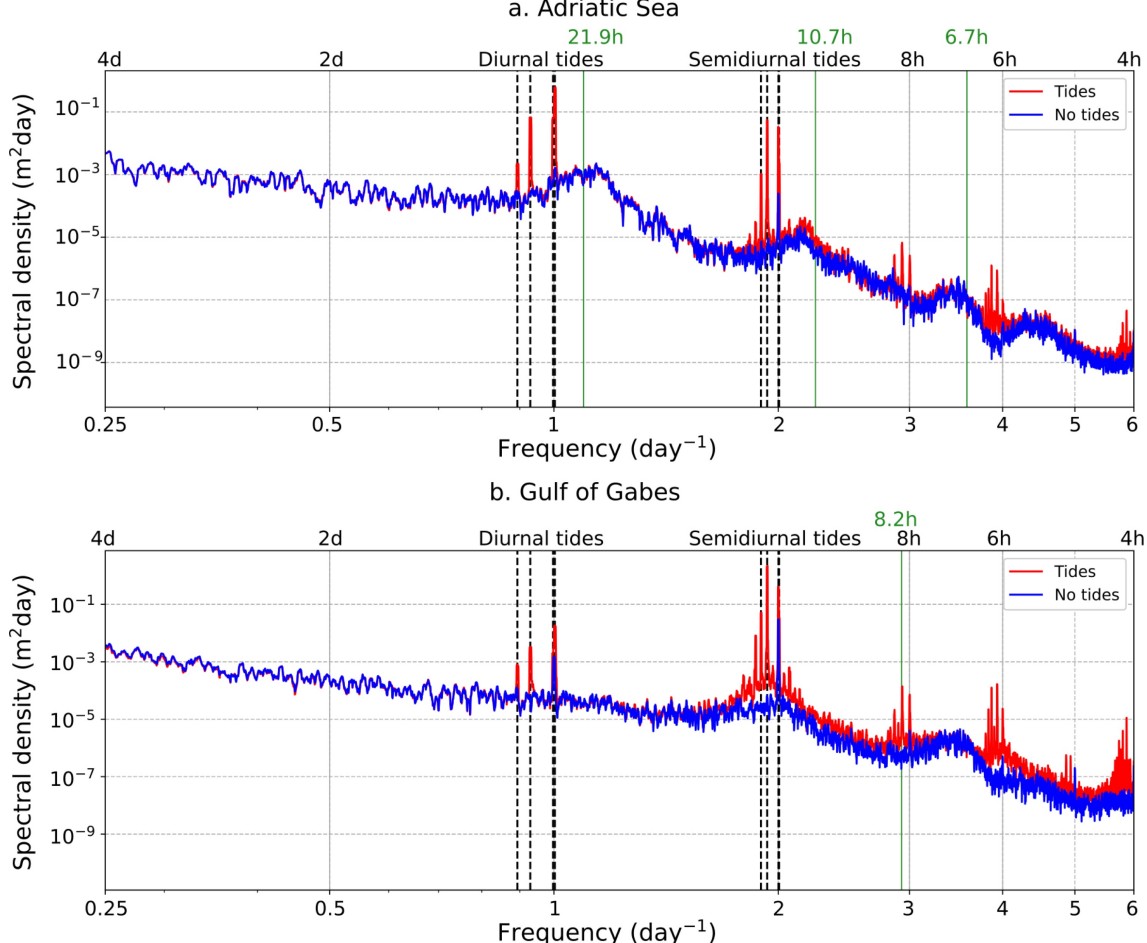

**Figure 3.** Energy density spectra of hourly mean sea surface height in the Mediterranean Sea for the period 2017-2021, for the model without tides (blue) and with tides (red), in a. Adriatic Sea (40.5°N-northern boundary), and b. Gulf of Gabes (32.5-35.5°N, 9.8-13.0°E). An area-weighted mean of the periodogram at each gridpoint was taken for each of the sub-basins. Green lines on the figure indicate frequencies of known barotropic oscillations in each region.

Considering the amplification of basin modes, Fig. 3 shows sea level spectra in the Adriatic Sea and Gulf of Gabes, to understand whether the peaks in Fig. 2 correspond to modes in these regions. In the Adriatic Sea (Fig. 3a), the sea level energy peaks at the frequencies of the barotropic modes of the Adriatic Sea at 10.7 hours and 12 hours (Lozano and Candela, 1995) are
enhanced by tides, whereas the mode at 6.7 hours (Schwab and Rao, 1983) has a similar energy density in both experiments. This is evidenced by the different amplitudes of the 10.7 and 12-hour peaks in the non-tidal case compared to the tidal one, unlike the 6.7-hour peak which remains the same. The In the Gulf of Gabes (Fig. 3b), tides also enhance the mode at 8.2 hours (Lozano and Candela, 1995).

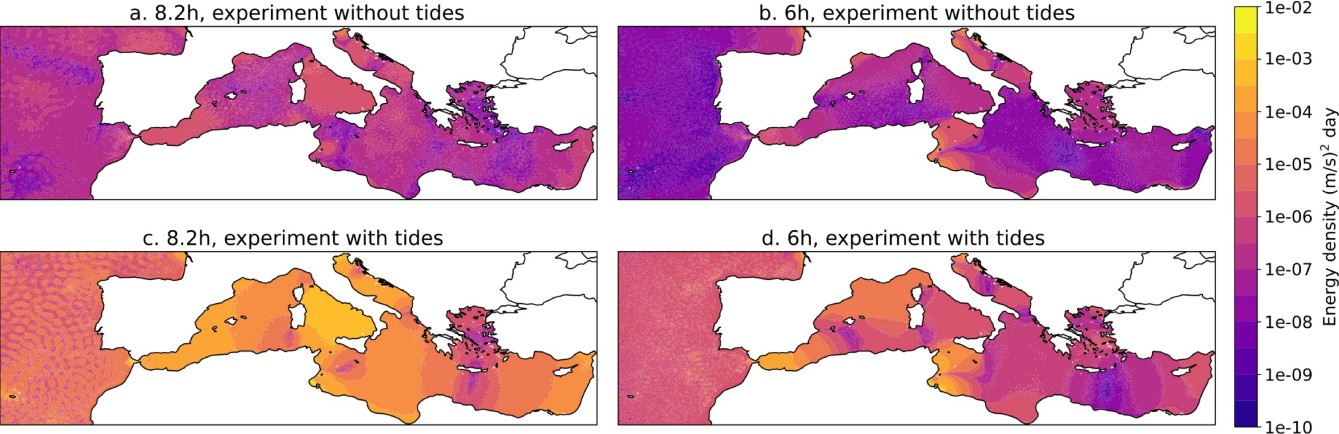

**Figure 4.** Maps of energy density of sea surface height at a. 8.2h without tides, b. 6h without tides, c. 8.2h with tides, and d. 6h with tides, for the Mediterranean Sea in 2017-2021.

Maps of the sea surface height power spectrum at 8.2h and 6h are shown for both experiments in Fig. 4. These frequencies are presented here as they show the largest differences between the two experiments out of the referenced frequencies from the literature (see Section 1), other than the tidal frequencies directly. Most of the Mediterranean Sea has enhanced power at 8.2h in the tidal experiment (Fig. 4a, c), with particularly large changes in the central Mediterranean: the Tyrrhenian Sea, the Gulf of Gabes, and the Adriatic Sea. The barotropic oscillation at 8.2h is in the Gulf of Gabes, according to Lozano and Candela (1995), but the third mode of the Mediterranean Sea (Schwab and Rao, 1983), and the nonlinear effects of tides in the central Mediterranean Sea according to Palma et al. (2020) are at frequencies close to this (8.4h and 8.0h respectively), and may also have an impact on Figure 4c. Since many of the calculations of barotropic oscillations in the Mediterranean Sea were made several decades ago, there is a need for an updated confirmation of the frequencies of barotropic oscillations using state-of-the-art methods. In Fig. 4b and 4d, we see that the Sicily escarpment region of Palma et al. (2020) again has particularly enhanced power in the experiment with tides, but other regions such as the Alboran Sea and the Western Mediterranean Sea see an interaction between tides and potential energy at the 6h frequency.

To summarise the analysis of sea surface height, we find that (1) tides affect the sea surface height on spatial and temporal scales away from those of the tides directly, (2) existing Mediterranean basin and regional barotropic oscillations at frequencies with a shorter time period than 12h are excited by tides, particularly at 6h and at several frequencies close to 8h, and (3) maps of the energy density at these frequencies reveal the distribution of these interactions in the Mediterranean Sea.

## 4 Kinetic energy spectra

For this analysis, several points are selected across the Mediterranean Sea, to understand the effects of tides on local phenomena such as the generation of internal tides and interactions with near-inertial waves. The map in Fig. 1 shows these points. From

this, three contrasting points were selected to be shown in this work: the Gibraltar Strait, the Tyrrhenian Sea, and the Cretan Sea. Results for the other points are available in the supplementary material.

We first calculate the rotary spectra for depth-averaged (barotropic) horizontal velocities (Fig. 5), for both clockwise and counter-clockwise components and then combine these to create the rotary kinetic energy density spectra. These rotary spectra visualise the time scales at which there is high kinetic energy at each selected point, over the entire water column. In the Gibraltar Strait (Fig. 5a), tides enhance kinetic energy at all frequencies, particularly at frequencies close to and higher than 12 hours. The basin mode frequency of 8.4 hours is enhanced by tides, and a peak at around 6h is also visible, which is due to nonlinear tidal effects (Palma et al., 2020). The Tyrrhenian Sea and Cretan Sea, unlike the Gibraltar Strait, have broad peaks at the near-inertial frequencies. Peaks at diurnal tidal frequencies are more apparent in the Tyrrhenian Sea than in the Cretan Sea.

We also analysed the kinetic energy spectra split into vertical levels, to consider baroclinic currents and internal wave modes, as shown in Fig. 6. We note that upper layer currents (0-150m) are characterised by the entering Atlantic Water layer dynamics, while the intermediate layer (150-500m) currents are on average in the opposite direction to the surface, characterising the Intermediate Water circulation in the basin. We now analyse the effects of tides on this anti-estuarine zonally oriented conveyor belt described in Pinardi et al. (2019). Figure 6 shows spectra for the three selected points throughout the water column.

In the Gibraltar Strait, it is clear that the entire structure of the kinetic energy is dominated by tides: tides enhance kinetic energy at almost all frequencies and depths (Fig. 6c). There are also internal tides visible in this region: Figure 6b shows peaks in kinetic energy at around 100m at the M2 and K1 tidal frequencies, implying the existence of internal tides in the Gibraltar Strait, as shown by Morozov et al. (2002), and more recently discussed in Gonzalez et al. (2023). Some diurnal internal tidal energy is visible in the Tyrrhenian Sea (Fig. 6e), as well as a shift in the near-inertial peak when compared to the experiment without tides (Fig. 6d). The interaction of tides with near-inertial waves is clearer in the Cretan Sea (Fig. 6i), as the peak is shifted to a higher frequency compared to Fig. 6e.

Internal tides are known to have large ageostrophic vertical velocities (Niwa and Hibiya, 2001; Li and von Storch, 2020) and thus an analysis of the vertical motion is mandatory. To visualise the internal tidal motion in the basin, we use Hovmoller diagrams for one month (May 2019) of hourly zonal currents for each of the three selected points from Fig. 6. In Fig. 7 the Gibraltar Strait location has a dominant vertical velocity oscillation at 12h. Contrastingly, inertial waves have greater importance in the Cretan Sea (Fig. 9) and they propagate downwards through the water column, notably around day 10-15 of the month and at 150-500m (Fig. 9c). The effect of tides on this is shown in Fig. 9d, where interactions between the internal tidal and near-inertial waves create wave-wave patterns during the same period. The Tyrrhenian Sea (Fig. 8) also has both near-inertial waves and internal tides at these depths, but the interaction here is weaker.

An interesting feature of Figs. 7-9, is the baroclinic structure of the vertical velocity in the experiment with and without tides. Two zero crossings appear, one approximately at 150m, the lower limit of the inflowing branch of the zonal conveyor belt already described above (Pinardi et al., 2019), and the second at 300m. This is particularly apparent in the Gibraltar Strait (Fig. 7), where internal tide generation leads to increased tidal velocity in both directions. The continuation of this zero-crossing at 150m in other regions (Figs. 8-9) demonstrates the importance of internal tide generation at the Gibraltar Strait and how it affects the entire Mediterranean Sea general circulation, including in remote regions.

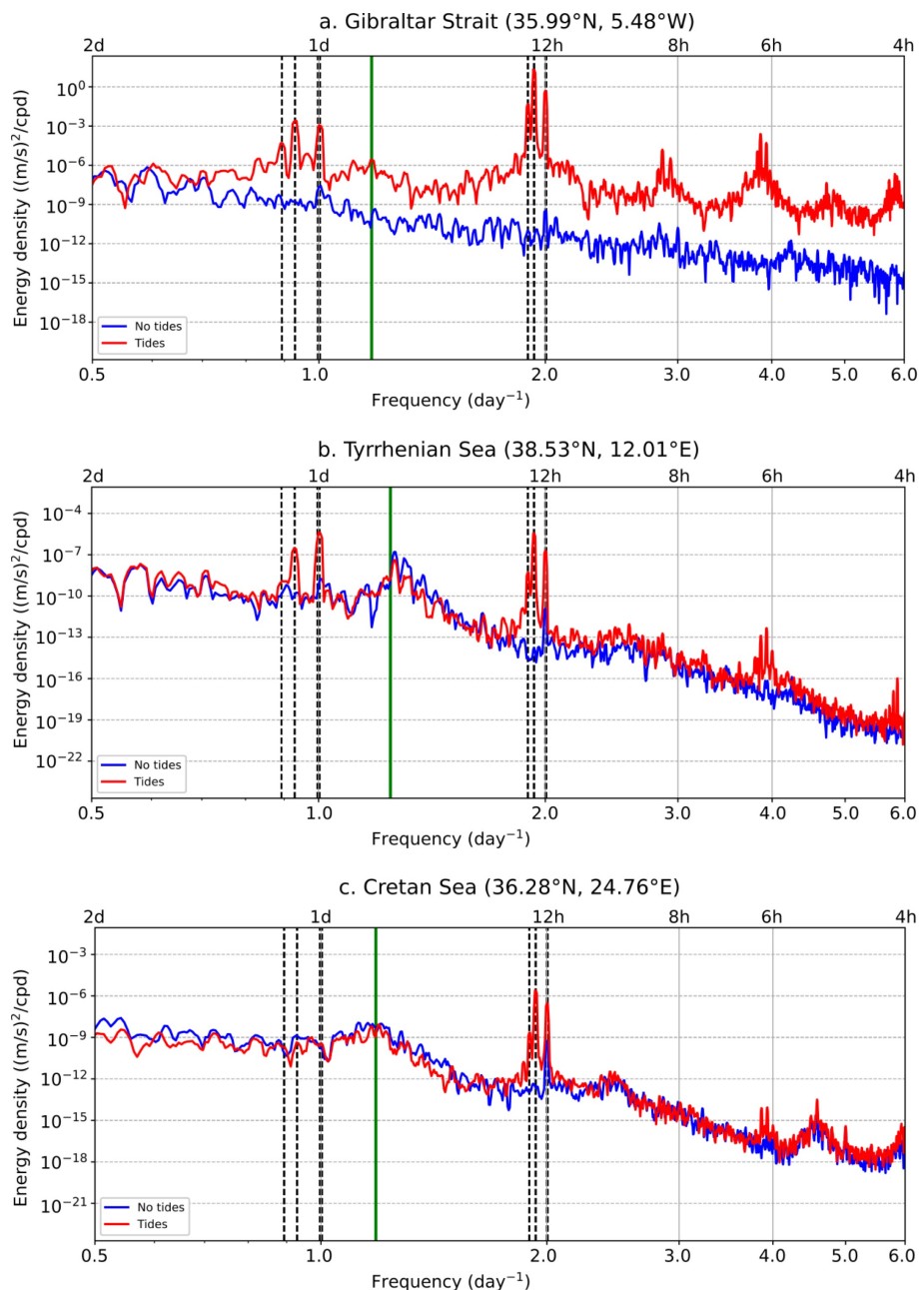

**Figure 5.** Barotropic (depth-mean) rotary kinetic energy density spectra for points in a. Gibraltar Strait (35.99°N, 5.48°W), b. Tyrrhenian Sea (38.53°N, 12.01°E), and c. Cretan Sea (36.28°N, 24.76°E), with the tidal experiment in red and the experiment without tides in blue, using hourly outputs over six months, January-June 2019. Dashed lines represent the eight tidal components used in the model, and the green line is the inertial frequency. The locations of these points are shown in Figure 1.

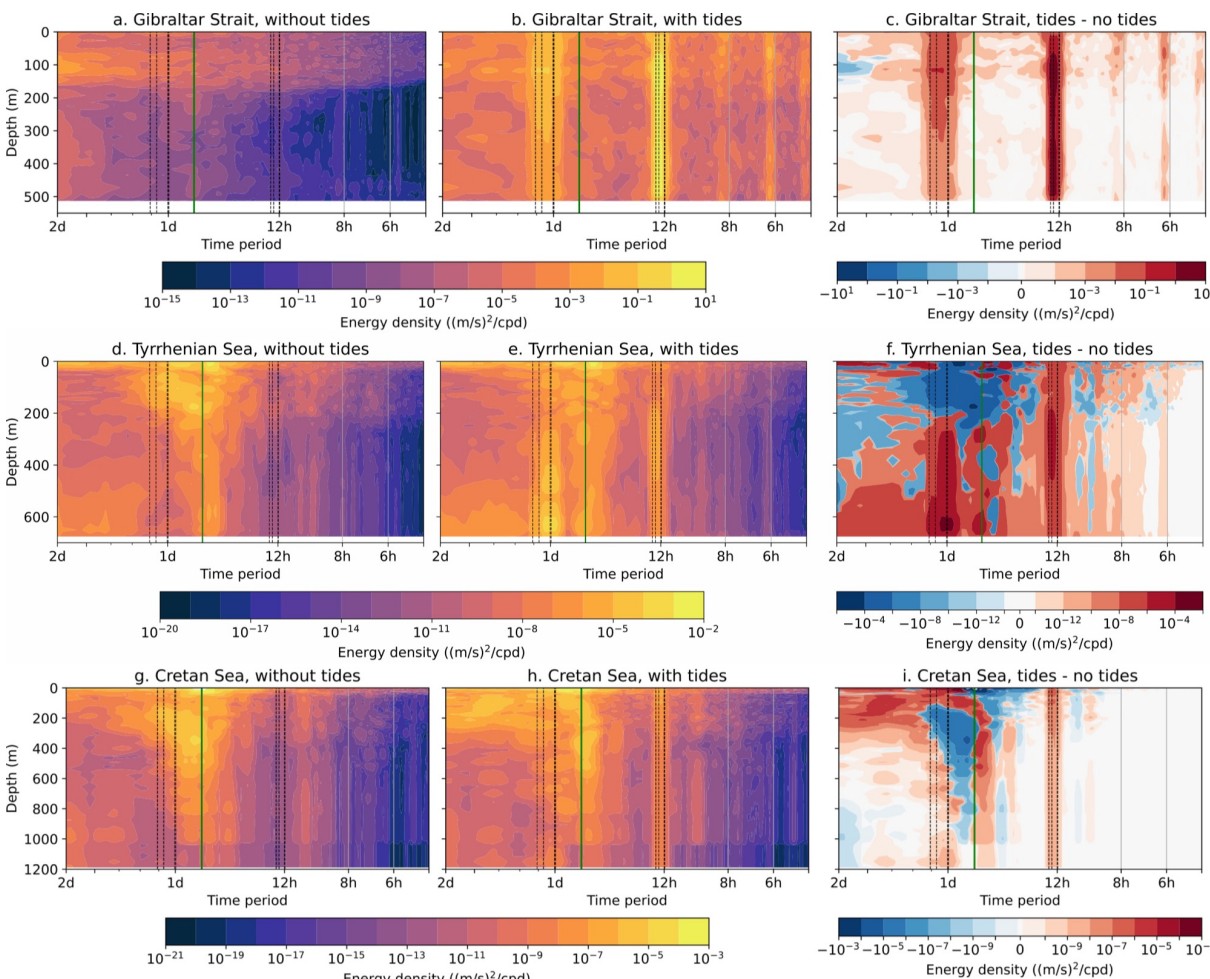

**Figure 6.** Rotary spectra of kinetic energy density over the full water column without tides at a. Gibraltar Strait (35.99°N, 5.48°W), d. Tyrrhenian Sea (38.53°N, 12.01°E), and g. Cretan Sea (36.28°N, 24.76°E), with tides at b. Gibraltar Strait, e. Tyrrhenian Sea, and h. Cretan Sea, and the difference between the two spectra at c. Gibraltar Strait, f. Tyrrhenian Sea, and i. Cretan Sea. All data are for May 2019. The locations of these points are shown in Figure 1. The green line represents the inertial frequency at each point, and the dashed lines indicate the frequencies of the eight tidal components included in the tidal experiment.

The three contrasting regions of the Mediterranean basin highlight the varying importance of tides, internal tides, and their interactions with near-inertial internal waves in different regions. While the Gibraltar Strait (Fig. 7) is dominated by semidiurnal 220   tides, the other two regions (Figs. 8-9) show propagating near-inertial waves, as in (Cozzani, 2023), interacting with internal tides, particularly in the Cretan Sea.

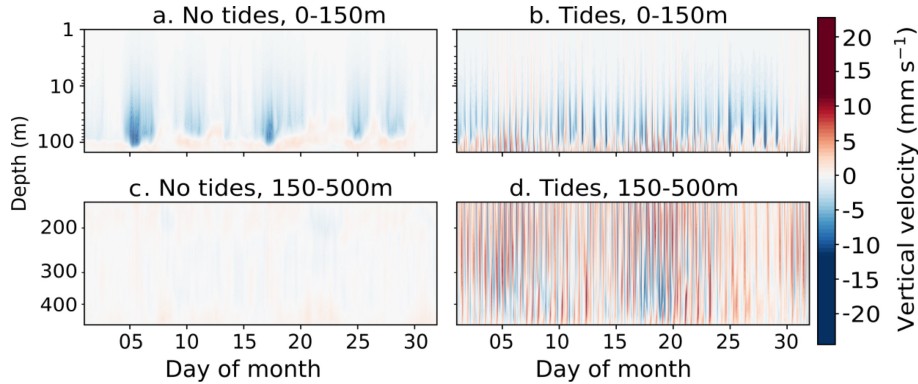

**Figure 7.** Hovmoller diagrams of depth against time of hourly mean vertical velocity at a point in the Gibraltar Strait (35.99°N, 5.48°W, see Figure 1) in May 2019, for a. Model without tides, 0-150m, b. Tidal model, 0-150m, c. Model without tides, 150-500m, and d. Tidal model, 150-500m. Note that the depth scale is logarithmic.

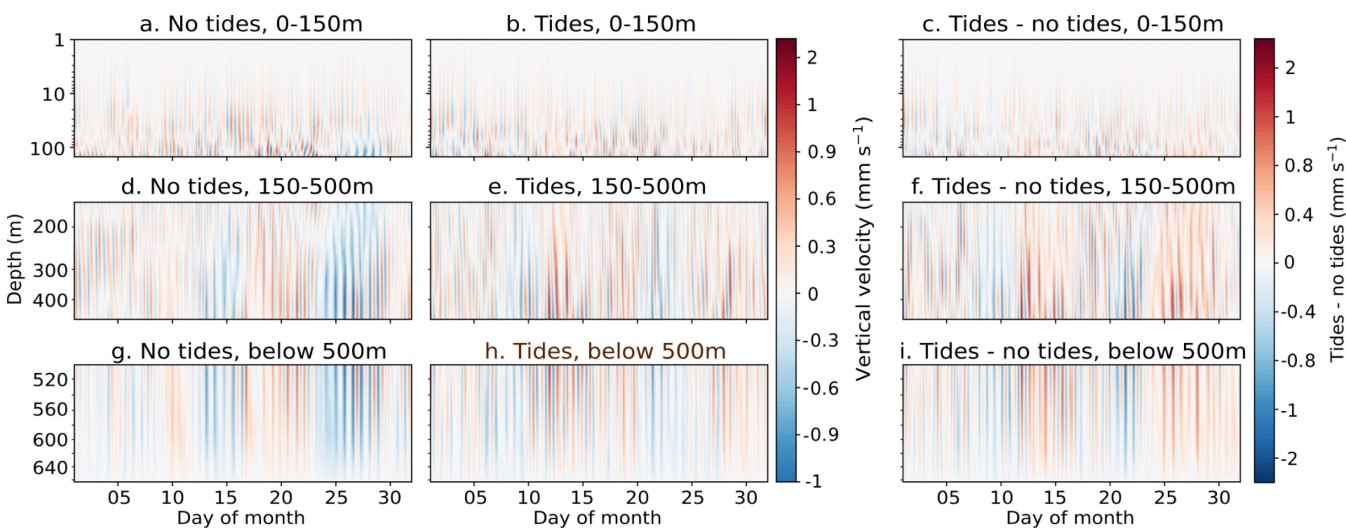

**Figure 8.** Hovmoller diagrams of depth against time of hourly mean vertical velocity at a point in the Tyrrhenian Sea (38.53°N, 12.01°E, see Figure 1) in May 2019, for a. Model without tides, 0-150m, b. Tidal model, 0-150m, c. Tidal model – model without tides, 0-150m, d. Model without tides, 150-500m, e. Tidal model, 150-500m, f. Tidal model – model without tides, g. Model without tides, below 500m, h. Tidal model, below 500m, and i. Tidal model – model without tides, below 500m. Note that the depth scale is logarithmic.

## 5    Mixed layer depth and water mass formation

Assessing the impact of tides on the mixed layer depth can provide indirect evidence for the impact of internal tides on the vertical mixing and vertical motion. The mixed layer depth is calculated via a density criterion based on a density change of 0.01 kgm$^{-3}$ (de Boyer Montégut et al., 2004). Figure 10 shows the change in winter (December-March) mixed layer depth in

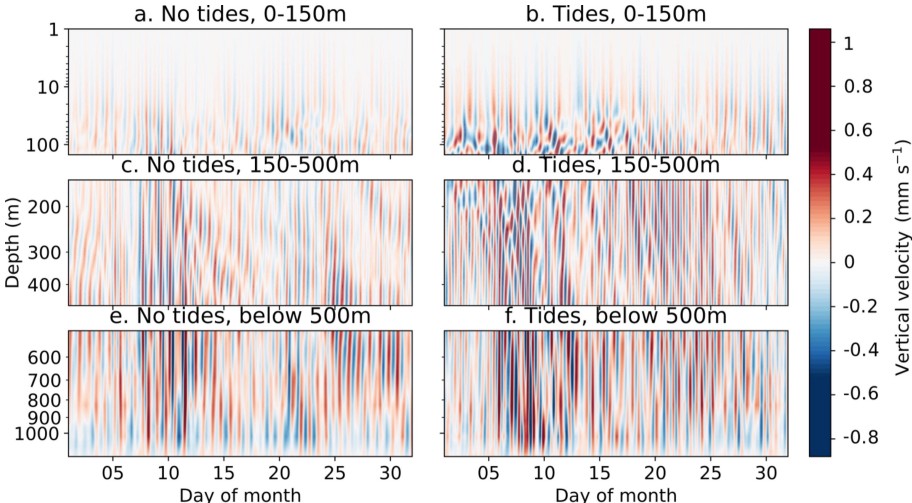

**Figure 9.** Hovmoller diagrams of depth against time of hourly mean vertical velocity at a point in the Cretan Sea (36.28°N, 24.76°E, see Figure 1) in May 2019, for a. Model without tides, 0-150m, b. Tidal model, 0-150m, c. Model without tides, 150-500m, d. Tidal model, 150-500m, e. Model without tides, below 500m, and f. Tidal model, below 500m. Note that the depth scale is logarithmic.

the two experiments, and a timeseries of the Mediterranean Sea mean mixed layer depth for the entire five-year period. There is an increase in winter mixed layer depth with tides throughout most of the basin, with notable exceptions in the Gibraltar Strait/Alboran Sea region, and in parts of the Aegean Sea. In the Gibraltar Strait, this is explained by a thicker interface layer (García-Lafuente et al., 2013), which shrinks the upper layer compared to the experiment without tides, reducing the depth at which the threshold density change for the mixed layer is reached. These effects extend into the upper-layer Alboran Sea water which originates in the Gibraltar Strait. In the Aegean Sea, despite having low amplitude barotropic tides, internal tides are present (Alford et al., 2012), and affect mixing at the bottom (Gregg et al., 2012), which likely in turn affects the water column structure in this shallow region.

However, the biggest change occurs in the Gulf of Lion region, a key area for the formation of deep water (Western Mediterranean Deep Water, WMDW, Fig. 1). Regarding the seasonality of mixed layer depth (Fig. 10 d-e), both the absolute and percentage change due to tides is positive throughout the year, but is greatest in winter. We also note that the largest increase in the mean mixed layer depth in the tidal experiment is evident during winter 2019, which is also the year characterised by the largest WMDW formation (see Fig. 11). The deepening of the mixed layer can be a precursor for increased dense water formation, but a direct analysis of this would be needed to confirm whether tides are enhancing water mass formation as well as mixed layer depth, and we note that the analysis in this work does not directly establish a connection between the two processes. Other important processes such as weak stratification and localised currents are key precursors to deep water formation in the Western Mediterranean.

Pinardi et al. (2015) defined that intermediate water has densities of 29.1-29.2 $kgm^{-3}$ and deep water has densities greater than 29.2 $kgm^{-3}$. Figure 11 shows the water mass formation rate in the deep and intermediate formation areas of Fig. 1, which

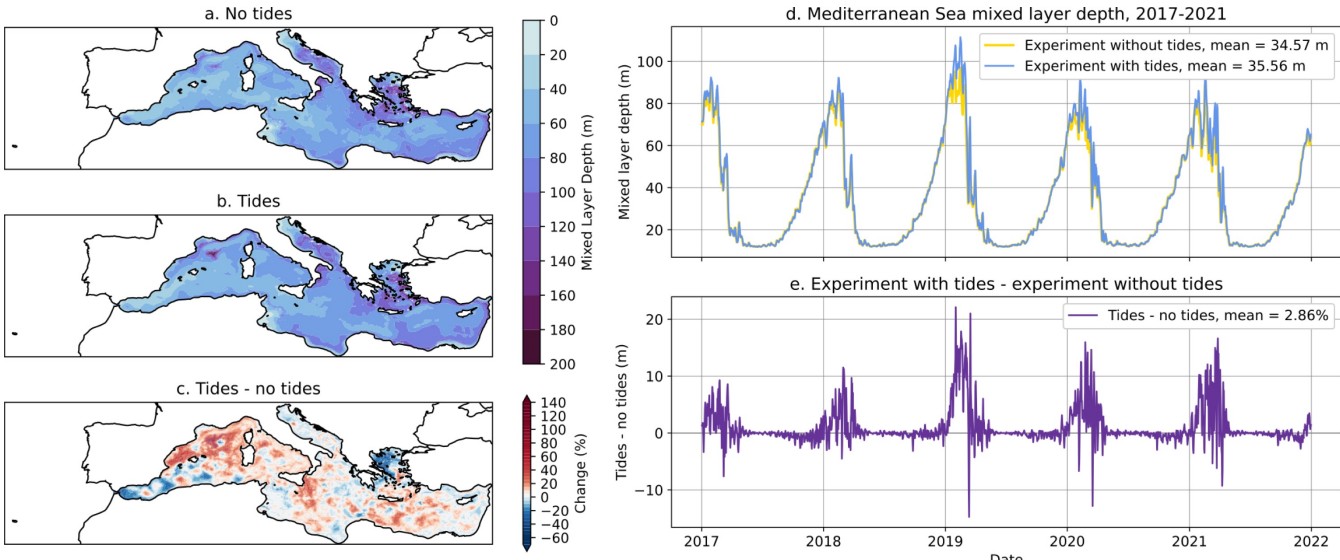

**Figure 10.** Winter (December-March) mean mixed layer depth in the Mediterranean Sea, for a. experiment without tides, b. experiment with tides, and c. the percentage difference between the tidal and non-tidal experiments. Timeseries of daily Mediterranean Sea mean mixed layer depth for 2017-2021 (d), and difference between tidal and non-tidal experiment timeseries (e).

are the same regions indicated by Pinardi et al. (2015), for each winter of the analysed period, with and without tides. The most notable change is in WMDW, particularly during the event of 2019, where the tidal experiment forms several times more deep water at its peak compared to the experiment without tides. This is in agreement with Naranjo et al. (2014), who showed that enhanced WMDW formation occurred with tides in four out of the five years analysed. Modest increases are also seen in LIW formation. EMDW and LDW have increased water mass formation rate when including tides in some years, and decreased in 250 other years. We argue that this irregular behaviour is connected to the impact of tides on the strength of the pre-conditioning factors and the air-sea interaction heat fluxes that are affected by tides (Oddo et al., 2023).

## 6  Temperature and salinity

The salinity and temperature of the Mediterranean Sea are affected by tides, primarily in the upper layer above 150m, as can be seen in Figure 12. This is the layer affected by the entering low salinity Atlantic water. As indicated in work by Naranjo 255 et al. (2014), Harzallah et al. (2014), Sanchez-Roman et al. (2018), and Gonzalez et al. (2023), inflowing salinity at Gibraltar increases when tides are introduced and upper layer temperature decreases, albeit by a smaller amount. This salinity increase has an upward trend that in time does not stabilise in the few years of our experiment. This was also found by Harzallah et al. (2014), where the salinity difference between the experiments with and without tides increased for several decades before stabilising, since this stabilisation depends on the overturning time scales of the basin that require several decades to spin up in 260 full.

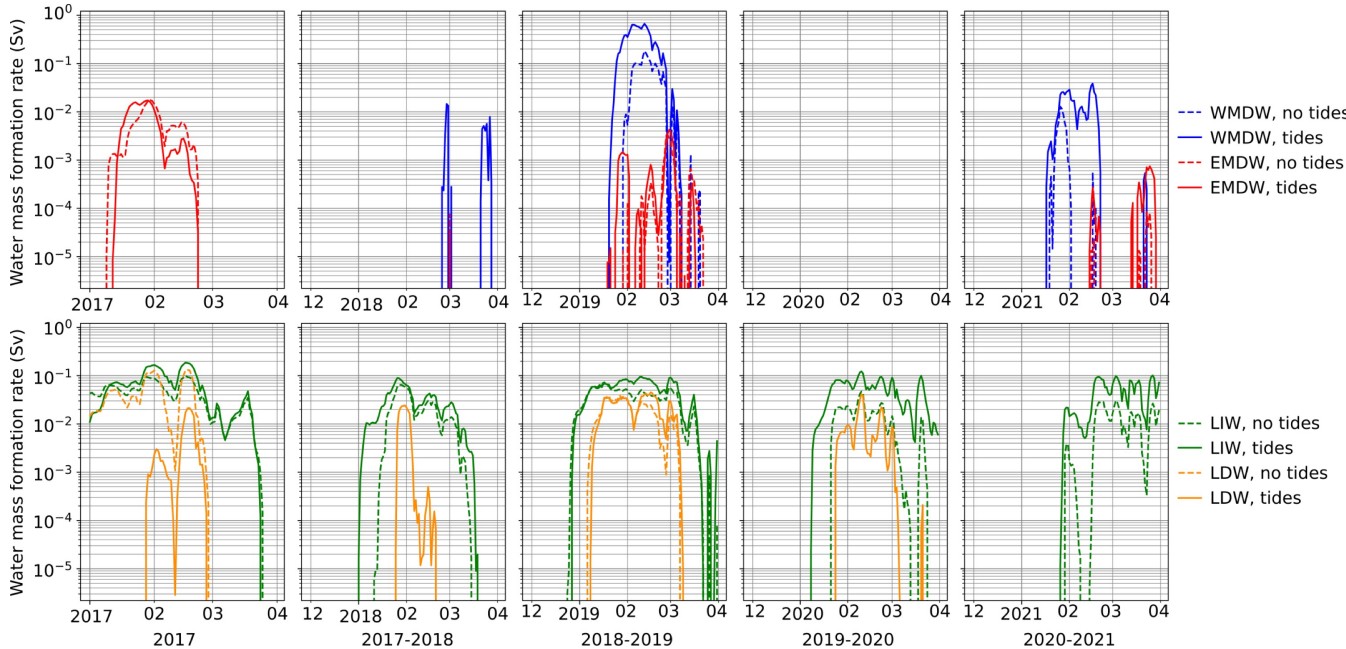

**Figure 11.** Daily water mass formation rate in winte (DJFM) 2017-2021, for the tidal experiment (solid lines) and the non-tidal experiment (dashed lines). Four types of water mass are included: Western Mediterranean Deep Water (blue), Eastern Mediterranean Deep Water (red), Levantine Intermediate Water (green), and Levantine Deep Water (orange).

The additional increased salinity in the Mediterranean Sea due to tides, and its trend over time, can be compared to observations from profiling floats to understand whether the salinity increase follows the observational trend or whether it is overestimated. This was calculated for the Mediterranean Sea and integrated along vertical layers for each year in the experiment. Figure 13 shows the mean salinity Root Mean Square Error (RMSE) of the experiments with respect to observations averaged in the whole basin and provided along nine vertical layers. For the tidal experiment, the RMSE does not present a clear linear trend, instead varying by individual year. The lack of linear trend is also apparent in the non-tidal experiment, but here the salinity bias is positive throughout the upper layer. Overall the RMSE of the two experiments is similar in some years, but in 2019 and 2020, the tidal experiment has a much lower salinity RMSE in the upper layers. Figure 13 demonstrates that in both experiments the model error is larger at the surface up to 150m and is reduced to less than 0.15 PSU below this layer, where it also shows a lower temporal variability. Lower layers remain stable with small errors, whereas the surface varies more and produces larger errors when compared to observations.

## 7 Conclusions

The effects of tides on the Mediterranean Sea circulation were studied through a twin numerical experiment approach based on a high-resolution (around 3.8 km) model of the region, with and without the explicit representation of tides. Spectra of sea

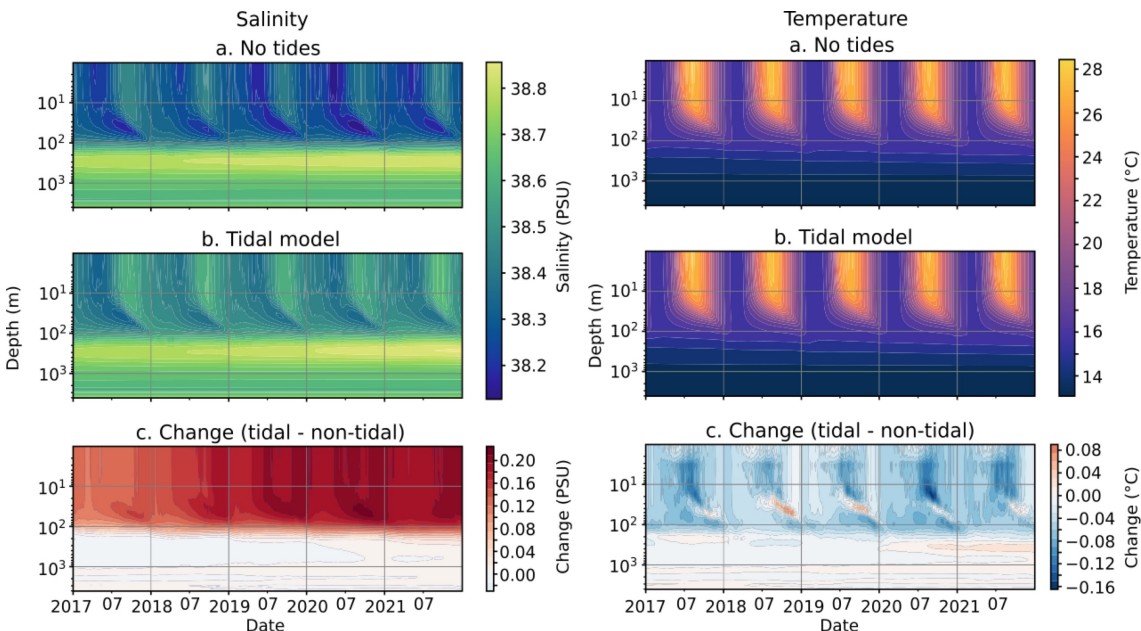

**Figure 12.** Hovmoller plot of depth against time of daily salinity (left) and temperature (right) in the Mediterranean Sea in 2017-2021, for the model without tides (a), with tides (b), and change (c): tidal model – model without tides. Note that the depth scale is logarithmic.

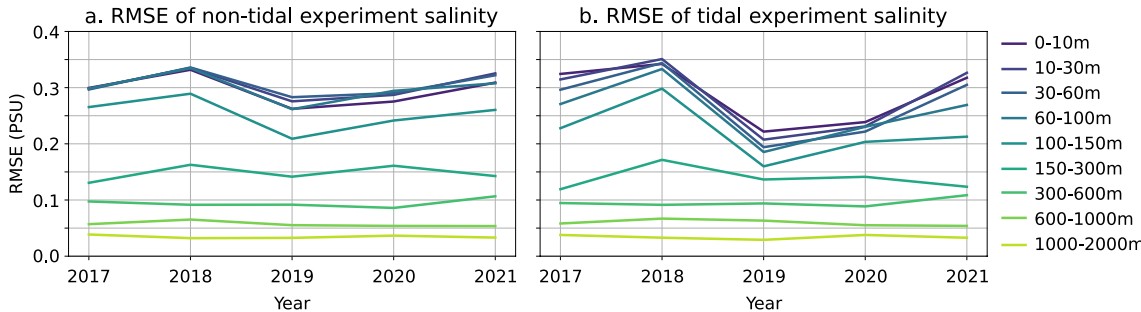

**Figure 13.** Salinity difference between the experiments and data from profiling floats for each analysed year (2017-2021), across nine layers for the Mediterranean Sea, presented as a. non-tidal experiment root mean square error, and b. tidal experiment root mean square error.

surface height and kinetic energy in the basin, as well as in key regions characterised by basin modes, revealed that the effects of tides on the energetics of the Mediterranean Sea extend far beyond the spatial and temporal scales of the tides themselves.

Tides enhance the power at 8h and 6h frequencies due to their nonlinear effects across the Mediterranean basin, amplifying the basin modes, as shown in the Sicily Strait by Palma et al. (2020). Tides also interact and amplify the other basin modes at 11.4h and 8.4h, as well as the Adriatic mode at 10.7h, and the mode in the Gulf of Gabes at 8.2h. The Adriatic modes at 12h and 21.9h are visible in both the tidal and non-tidal experiments, but the 12h frequency is enhanced by tides. Results such as these demonstrate the complexity and non-linearity of tidal effects in the Mediterranean Sea.

The study then discussed the rotary spectra of the barotropic and full kinetic energy in several selected points in the Mediterranean Sea. This analysis showed the ubiquitous field of internal waves below around 150 metres, both at diurnal and semidiurnal time frequencies. The Hovmoller diagrams of vertical velocity components revealed the internal tide field structure in the basin, as well as the interaction of internal tides with near-inertial waves in some regions, such as the Cretan Sea. In addition to near-inertial waves, both the semidiurnal and diurnal internal tides have high energy, potentially at an overall Mediterranean scale not previously highlighted.

Tides affect the mixed layer depth of the basin, with different signs in different regions. The Mediterranean deep and intermediate water formation is enhanced by tides in the WMDW region, and in the Levantine Sea where intermediate water masses are formed. This aligns with increases in the winter mixed layer depth across most of the Mediterranean Sea. It should be noted that although an increased mixed layer depth is one conditioning element for dense water formation, other concomitant processes are needed for deep water formation events to occur (Marshall and Schott, 1999; Pinardi et al., 2022).

It is important to note that the resolution of this model configuration, although higher than many global ocean models and regional ocean models from past studies, has limitations in its representation of waves with shorter wavelengths, including internal tides. According to Li et al. (2015, 2017), the first two modes of the M2 internal tide and first three modes of the K1 internal tide all have wavelengths greater than 45 km, but high modes are likely unresolved at the $\frac{1}{24}^{\circ}$ resolution. Furthermore, high vertical resolution is important for simulating internal tides. Our model lacks high vertical resolution at deeper levels, and this could contribute to a lack of internal tides below 500m.

In this work, the interactions between tides and other high-frequency phenomena were described, but a longer model integration, such as that of Pinardi et al. (2019) but including tides, would reveal the effects of tides on lower-frequency phenomena. Moreover, a dedicated study of internal tides, as has been carried out in the global ocean by Li et al. (2015, 2017), in the Mediterranean Sea is lacking and our results provide further motivation for this. The current literature also lacks a calculation of the barotropic modes of the Mediterranean Sea using state-of-the-art techniques which would provide an important dataset for further studies on the barotropic oscillations in the Mediterranean Sea.

*Data availability.* The datasets used for the analysis done in this work can be found at https://doi.org/10.5281/zenodo.10911630.

*Author contributions.* BM ran the numerical models, analysed the results, and wrote the manuscript. EC and AG provided support in the set-up and running of the numerical models and analysis. NP planned the study and supported the analysis of model results. All authors reviewed and edited the manuscript.

*Competing interests.* The authors declare that they have no conflicts of interest.

*Acknowledgements.* We would like to thank two anonymous reviewers, and our editor, Joanne Williams, for their feedback that led to the improvement of this work. This study has been conducted including support from the University of Bologna Ph.D. programme in Future Earth, Climate Change, and Societal Challenge, and the EU Copernicus Marine Service for the Mediterranean Monitoring and Forecasting Center.

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
