# Peer review of "The characteristics of tides and their effects on the general circulation of the Mediterranean Sea"

_EGUsphere, 2023_

## Referee Comment (RC1)

**Review of "The characteristics of tides and their effects on the general circulation of the Mediterranean Sea" by McDonagh et al.**

The manuscript "The characteristics of tides and their effects on the general circulation of the Mediterranean Sea" by McDonagh et al. presents a study investigating the effect of tides on the Mediterranean Sea circulation. To my knowledge, this subject has not yet been extensively investigated. Apart from Sannino et al. 2014, which provided a first analysis of the tidal influence on the large-scale Mediterranean circulation, previous studies have only focused on specific areas such as the Alboran Sea (Sanchez-Garrido et al., 2013) or the Sicily Strait (Oddo et al., 2023; Gasparini et al., 2004). As such, the present manuscript proposes a valuable contribution to reinforce and deepen the current knowledge on the tidal influence on the Mediterranean Sea.

In this study, the authors diagnose the effect of tides from a pair of 5-year-long tidal and non-tidal experiments based on very similar numerical configurations. The first and second sections of the manuscript investigate the influence of tides on the Mediterranean Sea dynamics through the prism of sea level and kinetic energy spectra. These analyses focus on three specific locations: the Strait of Gibraltar, the Tyrrhenian Sea, and the Cretan Sea. They reveal that tides impact high-frequency (> 1 day$^{-1}$) dynamics through the propagation of the main tidal harmonics at work within the Mediterranean Basin and their interaction with various basin-scale modes. Then, the authors relate the tidally enhanced dynamics and mixing to the deepening of the mixed layer depth in the tidal simulation. The two final sections discuss the impact of tides on the thermohaline properties of the Mediterranean Sea and the transports through the Strait of Gibraltar.

Overall, the manuscript has the potential to provide valuable results on tidal contribution to Mediterranean dynamics. However, further work must be done before it can be accepted. More specifically, I think that although the two first sections of the manuscript provide valuable results, they focus on too specific areas to provide an overall picture of the tidal influence on the high-frequency Mediterranean dynamics. In addition, to assess the influence of tides on the "general circulation of the Mediterranean Sea", as stated in the the title of the study, the manuscript should investigate the influence of tides on the long-term, large-scale circulation. Regarding the impact of tidal dynamics, section 5 should more clearly distinguish the influence of local tidal mixing at the strait of Gibraltar, which impacts the Mediterranean mixed layer depth indirectly by changing the density of Atlantic water masses, and the less intense mixing induced by tidal currents throughout the Mediterranean Sea. The former is not directly related to the interaction of tides with the Mediterranean circulation, and it has already been investigated with similar model configurations (Sannino et al., 2014; Naranjo et al., 2014). Thus, I suggest not including it in this manuscript. On the other hand, the mixing induced by tidal dynamics is relevant to this study. Finally, sections 6 and 7 are, in my opinion, outside of the scope of this paper. Although interesting, the corpus of these sections has no apparent link with Mediterranean circulation and mainly emphasizes the conclusions of previous studies without additional results.

For these reasons, I am arguing for a major revision of the manuscript. Specifically, I would suggest:

- In sections 3 and 4: Add 2D maps of the tidal influence over the specific frequency bands mentioned in the text to give further confidence in the spatial extension of the results discussed.
- In section 5: If you intend to show that the tide-enhanced high-frequency dynamics are responsible for the deepening of the Mediterranean mixed layer depth, you should:
  (1) Mostly focus on the overall increase of the mixed layer depth rather than its local increase over deep convection areas, where the intensity of tidal mixing is unlikely strong enough to drive the deepening.
  (2) Look at the seasonal cycle of the mixed layer depth and stratification, as it would be easier to separate the effect of local vertical mixing from that of the tide-induced densification of Atlantic water masses at the Strait of Gibraltar.
- Add some results on the influence of tides on the Mediterranean large-scale circulation, or reformulate the title of the article only to consider the high-frequency dynamics.
- Put sections 6 and 7 in the supplementary materials or a "model validation" section, demonstrating the consistency of the model with previous studies.

I do encourage authors to make the necessary effort to improve this manuscript. You can find general and detailed remarks below.

**General remarks:**
- There is no model validation. At least a reference to the model validation should be included.
- The tidal and non-tidal simulations differ by other processes than tides. Please provide some information about the impact of these differences. The best would be to briefly analyze these impacts in the supplementary materials.
- Text clarity: Please use as few indirect forms as possible to make the manuscript easier to read.

**Introduction:**

**General remarks:**
The introduction is relatively well documented. I thank the authors for this time-consuming work. However, it would benefit from a more straightforward structure. As it is now, the introduction paragraphs discuss:
1. Tides in numerical models and their relevance to large-scale circulation
2. Influence of tides at the Strait of Gibraltar.
3. Influence of internal tides in the Mediterranean Sea.
4. Description of internal tides and their influence on the global ocean and the Mediterranean Sea
5. Basin modes of the Mediterranean Sea

I suggest you start with a general introduction, including the content in paragraphs 1 and 4, to introduce the various aspects of tidal waves, their influence on large-scale circulation and mixing, and how they are represented. Then, explain how the tides influence the Mediterranean Sea, as in paragraphs 2, 3, and 5.

Also, you should further motivate the need for a deeper understanding of the effect of tidal motion on the Mediterranean circulation. In the current version of the manuscript, this is only mentioned in one sentence "Many of these free oscillations could be affected or enhanced by tides, especially considering their proximity to tidal frequencies."

**Detailed remarks:**
- "Tidal forcing is a rather recent addition to large scale circulation models that start to have horizontal and vertical resolutions that allow for an analysis of tidal motion on the circulation" => This sentence is unclear. Do you mean that horizontal and vertical resolutions are now fine enough to represent tides properly? Please clarify this.
- "Tides are now considered to be essential components of the large scale circulation" => You should provide references specifically investigating the influence of tides on the large-scale circulation, for example: Müller et al., 2010
- "Recently, Gonzalez (2023) has revisited the tidal dynamics in the Gibraltar Strait, concluding that there are several tidal-induced hydraulic control points and the authors developed a specific mixing parametrization for the Strait." => The Ph.D. of Gonzalez (2023) does not directly investigate hydraulic control points at the Strait of Gibraltar. These were observed by (Farmer & Armi, 1985; Farmer et al., 1988) and discussed by Brandt et al., 1996; Vázquez et al., 2006; Vlasenko et al., 2009; Sánchez-Román et al., 2012; García Lafuente et al., 2013; Hilt et al., 2020.
- "Harzallah et al. (2016) and Naranjo et al. (2014) found that tides at the Strait of Gibraltar: (1) increase the baroclinic volume transport, (2) increase the salinity of Atlantic inflowing waters through the enhancement of mixing, affecting the water mass formation processes further downstream from the Strait and (3) change the Mediterranean deep water outflow." =>
  (1) As I understand it, Harzallah et al. (2016) do not state that tides increase the baroclinic volume transport. In fact, we can read from the abstract of the paper: "It is shown that tidal oscillations reduce the two-way exchange by interaction with the subinertial variability.". The fact that tides increase the baroclinic volume transport is not so evident to me. The paper of Gonzalez et al. 2023 shows that the computed transports through the Strait depend highly on the chosen definition for the interface between inflowing and outflowing waters. Instead of stating that tides increase the baroclinic transports, I suggest that you say they intensify the high-frequency dynamics of the Strait.
  (2) Tides also cool the Atlantic water masses, although to a lesser extent than they salten it.
  (3) You should specify what properties of the Mediterranean deep outflow are changed.
- I cannot find van Haren et al. (2014) in the references. Please add it.
- I think there is a mistake in the reference of the paper of Harzallah et al. (2016), which was published in 2014.

**Data and methods:**

**Detailed remarks:**
- "Lateral open boundary conditions are used in the Atlantic Ocean and Dardanelles Strait (see Fig. 1)." => What dataset do you use to force the model at these boundaries? Please provide a reference.
- "Additionally, 70 monthly mean climatological freshwater inputs from 39 rivers are added to the surface layer." => Please provide a reference for this dataset.
- Please explain how you choose the constant values used in the vertical mixing parameterization.

**Sea level energy spectra:**

**General remarks:**
In this section, you should emphasize the innovative aspect of your work more clearly. One way to do so would be to discuss separately the interaction of tides with basin-scale frequencies highlighted in previous studies, which you confirm here, and those your study is the first to highlight. Also, a short recap on your findings and their impact at the end of the section would be welcome.

**Detailed remarks:**
- "energy is reduced in the tidal model at frequencies lower than 0.5 d−1 (longer than a period of two days)." => Please specify the frequency in days first, as done in the following.

**Kinetic energy spectra:**

**General remarks:**
- In my opinion, you should remove Figures 7-9 from the corpus of the manuscript. They are used only in a small paragraph, do not provide new information with respect to Figure 6, and focus on a relatively short period of 1 month.
- Figures 10-12 should be replaced by the equivalent of Figure 6 for the vertical velocities. This would make it easier to read the impact of tides on the frequency band mentioned in the text and make the results more robust, as they would integrate the 5 years of simulation instead of one month.

**Detailed remarks:**
- "We first calculate the rotary spectra for depth-averaged (barotropic) horizontal velocities" => Please include some explanation about the physical meaning of this spectra.
- "The spectra of kinetic energy density were split into vertical levels to consider baroclinic currents and internal wave modes. [...] characterising the Intermediate

Water circulation in the basin." => It is unclear which Figure you refer to. I assume it is Figure 6, but it is confusing since you introduce it later. If the figure you refer to is not in the manuscript, specify it with the mention "(not shown)".

- "implying the existence of internal tides as already shown by Gonzalez (2023)." => Here, you should refer to the paper of Gonzalez et al., 2023. Also, many studies investigated internal tides in the Strait of Gibraltar. It is consistent to cite Gonzalez et al. 2023 here, but if you want to cite it alone, you should write something like: "implying the existence of internal tides, recently highlighted in Gonzalez et al. (2023)."
- "Two zero crossings appear, one approximately at 150m, the lower limit of the inflowing branch of the zonal [...]" => Please detail the implications of this result.

**Mixed layer depth and water mass formation:**

**Detailed remarks:**
"There is an increase in mixed layer depth with tides throughout most of the basin, with notable exceptions in the Gibraltar Strait/Alboran Sea region, and in parts of the Aegean Sea." => Please explain why the mixed layer depth responds differently to tides in these regions.

**Temperature and salinity:**

**Detailed remarks:**
"As indicated in work by Naranjo et al. (2014) and Harzallah et al. (2016), inflowing salinity at Gibraltar increases when tides are introduced and upper layer temperature decreases" => You should also cite the papers of Gonzalez et al 2023, Sanchez-Roman et al., 2018, and Sannino et al 2014,

**The Gibraltar Strait:**

**General remarks:**
You should specify how you compute the inflow and outflow transports through the strait of Gibraltar. As discussed in Gonzalez et al. 2023, it significantly impacts the transport values obtained.

**Detailed remarks:**
"Gonzalez (2023) recently demonstrated that the required resolution for an accurate representation of the Gibraltar Strait would be about five times the one used in our model." => It is Sannino et al., 2009 who discuss the resolution needed to represent the Strait of Gibraltar, not Gonzalez et al., 2023.

**Figures:**

**Fig1:** Please adapt the colormap so it is possible to see the tidal amplitude in the Atlantic. I understand tidal amplitudes significantly differ over the Mediterranean and the Atlantic, making it difficult to plot both with one colormap. However, you could use one colormap for the Atlantic and one for the Mediterranean. Also, add the tidal phase in contours to display amphidromic points.

**Fig. 2:** Add the scale in days for the two lower panels. Please indicate the frequency you refer to in the text using dashed lines and display the labels of tidal harmonics next to the associated peaks.

**Fig. 3:** Please indicate the frequency you refer to in the text using dashed lines and display the labels of tidal harmonics next to the associated peaks.

**Fig. 4:** I do not find any reference to the bathymetry in the manuscript, so you should indicate the points specified here in Figure 1 and remove this figure.

**Fig. 6:** Please add a panel displaying the differences between the two simulations.

**References:**

Brandt P, Alpers W, Backhaus JO (1996) Study of the generation and propagation of internal waves in the Strait of Gibraltar using a numerical model and synthetic aperture radar images of the European ERS 1 satellite. Journal of Geophysical Research: Oceans 101(C6):14,237–14,252, DOI 10.1029/96JC00540, URL http://doi.wiley.com/10.1029/96JC00540

Farmer DM, Armi L (1985) The internal hydraulics of the Strait of Gibraltar and associated sills and narrows. Oceanologica acta 8(1):37–46, URL https://archimer.ifremer.fr/doc/00112/22325/

Farmer DM, Armi L, Armi L, Farmer DM (1988) The flow of Atlantic water through the Strait of Gibraltar. Progress in Oceanography 21(1):1, DOI 10.1016/0079-6611(88)90055-9, URL https://doi.org/10.1016/0079-6611(88)90055-9

Garcia Lafuente J, Bruque Pozas E, S´anchez Garrido JC, Sannino G, Sammartino S (2013) The interface mixing layer and the tidal dynamics at the eastern part of the Strait of Gibraltar. Journal of Marine Systems 117-118:31–42, DOI 10.1016/j.jmarsys.2013.02.014, URL http://dx.doi.org/10.1016/j.jmarsys.2013.02.014

Gasparini GP, Smeed DA, Alderson S, Sparnocchia S, Vetrano A, Mazzola S (2004) Tidal and subtidal currents in the Strait of Sicily. Journal of Geophysical Research: Oceans 109(2), DOI 10.1029/2003jc002011

Gonzalez N, Waldman R, Sannino G, Giordani H, Somot S (2023) Understanding tidal mixing at the Strait of Gibraltar: A high-resolution model approach. Progress in Oceanography 212:102,980, DOI 10.1016/j.pocean.2023.102980, URL https://linkinghub.elsevier.com/retrieve/pii/S007966112300023X

Harzallah A, Alioua M, Li L (2014) Mass exchange at the Strait of Gibraltar in response to tidal and lower frequency forcing as simulated by a Mediterranean Sea model. Tellus A: Dynamic Meteorology and Oceanography 66(1):23,871, DOI 10.3402/tellusa.v66.23871, URL https://doi.org/10.3402/tellusa.v66.23871

Hilt M, Auclair F, Benshila R, Bordois L, Capet X, Debreu L, Dumas F, Jullien S, Lemari´e F, Marchesiello P, Nguyen C, Roblou L (2020) Numerical modelling of hydraulic control, solitary waves and primary instabilities in the Strait of Gibraltar. Ocean Modelling 151(May):101,642, DOI 10.1016/j.ocemod.2020.101642, URL https://linkinghub.elsevier.com/retrieve/pii/S146350032030144X

Muller M, Haak H, Jungclaus JH, S¨undermann J, Thomas M (2010) The effect of ocean tides on a climate model simulation. Ocean Modelling 35(4):304–313, DOI 10.1016/j.ocemod.2010.09.001, URL https://linkinghub.elsevier.com/retrieve/pii/S1463500310001289

Naranjo C, Garcia-Lafuente J, Sannino G, Sanchez-Garrido JC (2014) How much do tides affect the circulation of the Mediterranean Sea? From local processes in the Strait of Gibraltar to basin-scale effects. Progress in Oceanography 127:108–116, DOI 10.1016/j.pocean.2014.06.005

Oddo P, Poulain PM, Falchetti S, Storto A, Zappa G (2023) Internal tides in the central Mediterranean Sea: observational evidence and numerical studies. Ocean Dynamics 73(3-4):145–163, DOI 10.1007/s10236-023-01545-z, URL https://doi.org/10.1007/s10236-023-01545-z https://link.springer.com/10.1007/s10236-023-01545-z

Sanchez-Garrido JC, Garc´ıa Lafuente J, ´Alvarez Fanjul E, Sotillo MG, de los Santos FJ (2013) What does cause the collapse of the western alboran gyre? results of an operational ocean model. Progress in Oceanography 116:142–153, DOI 10.1016/j.pocean.2013.07.002, URL http://dx.doi.org/10.1016/j.pocean.2013.07.002

Sanchez-Román A, García-Lafuente J, Delgado J, S´anchez-Garrido JC, Naranjo C (2012) Spatial and temporal variability of tidal flow in the Strait of Gibraltar. Journal of Marine Systems 98-99:9–17, DOI 10.1016/j.jmarsys.2012.02.011, URL http://dx.doi.org/10.1016/j.jmarsys.2012.02.011

Sanchez-Roman A, Jorda G, Sannino G, Gomis D (2018) Modelling study of transformations of the exchange flows along the Strait of Gibraltar. Ocean Science 14(6):1547–1566, DOI 10.5194/os-14-1547-2018

Sannino G, Garrido JC, Liberti L, Pratt L (2014) Exchange Flow through the Strait of Gibraltar as Simulated by a σ-Coordinate Hydrostatic Model and a z-Coordinate Nonhydrostatic Model. The Mediterranean Sea: Temporal Variability and Spatial Patterns 9781118847:25–50, DOI 10.1002/9781118847572.ch3

Vazquez A, Stashchuk N, Vlasenko V, Bruno M, Izquierdo A, Gallacher PC (2006) Evidence of multimodal structure of the baroclinic tide in the Strait of Gibraltar. Geophysical Research Letters 33(17):L17,605, DOI 10.1029/2006GL026806, URL http://doi.wiley.com/10.1029/2006GL026806

Vlasenko V, Sanchez Garrido JC, Stashchuk N, Lafuente JG, Losada M (2009) Three-dimensional evolution of large-amplitude internal waves in the Strait of Gibraltar. Journal of Physical Oceanography 39(9):2230–2246, DOI 10.1175/2009JPO4007.1

---

## Referee Comment (RC2)

**Review of: "The characteristics of tides and their effects on the general circulation of the Mediterranean Sea"**

By *Bethany McDonagh, Emanuela Clementi, Anna Chiara Goglio, and Nadia Pinardi*.

https://doi.org/10.5194/egusphere-2023-2251

**Overview of the paper:**

This paper undertakes an assessment on a basin scale of the effects of tides on the Mediterranean Sea's circulation. The methodology involves an analysis of two experiments, one with tides and one without. It should be noted that tides aside, the experiments do have some other differences. To my knowledge there has not been such a systematic analysis of the effect of tides to the Mediterranean basin as a whole and expands on several studies which focus on specific sub regions. Thus, the subject matter of this paper is both very relevant to the journal and makes a substantial contribution beyond what has been done to date.

The twin experiments are of 5 years duration. To understand the effect on the Mediterranean dynamics the first parts of the paper focus on sea level and kinetic spectra in the following locations, the Strait of Gibraltar, the Tyrrhenian Sea, and the Cretan Sea. Clear Amplification of several sub daily modes is demonstrated by the analysis. The next section goes on to investigate how this tidal amplification leads to deepening of the mixed layer depth. To finish the paper discusses the effects of the tides on the thermohaline structure of the Mediterranean and briefly the baroclinic transport through the Strait of Gibraltar.

**General constructive critiques:**

The sections on the energy spectra are in the end limited to specific locations of interest. I can see why this is done as a pragmatic decision, but it lessens the impact of the paper which is aimed at a more general analysis of impact to the basin scale circulation. Perhaps a broader scale analysis in complement to the site-specific analysis could be done to shore up the results here and conclusions inferred from them such as broader scale maps?

It might be worth raising limitations of the model at only 4km resolution with a hydrostatic model to faithfully represent internal tides. It wouldn't detract from the paper overall as the point is increased tidal energetics exciting existing modes. But of course, the accuracy of the internal tides themselves would be questionable in such a configuration and miss some key processes. The point here is to bring such limitations of the configuration to the attention of the reader so they do not over infer the models representation of internal tides.

The twin experiments do have some differences other than the tides. I understand that the differences will probably be very small compared to the tides themselves, but I think it is worth at least conducing as short simulation (order 1 month) of the no tides solution with the exact same options as the tide simulation to confirm that the other differences are negligible.

The final 2 sections are quite abrupt and may in the end not particularly conclusive in either case. That said "negative results" are also of value and the suggestion with regards to resolution for 7 is a

very reasonable further line of inquiry. I would not demand that the sections are removed I just feel they may not in their current form be adding anything substantial to the paper in its current form and may be better left in 8 as future lines of inquiry to investigate further.

**Section specific points**

(general tiny niggle, this is picky and trivial, but can the units and numbers be spaced out e.g. 3.5 TW instead of 3.5TW etc.)

**1 Introduction**

The introduction in my view does a good job of introducing the relevant background research. It might be worth mentioning the increasing importance of tides in coupled climate simulations, though the Arbic reference also goes into this, and perhaps that this is important for future simulations of the Mediterranean particularly at climate time scales if the water formation regions are affected.

**2 Data and Methods**

As mentioned above it might be nice to rule out any differences induced by the time stepping by a short run of the no tides simulation with the exact settings and bathy as used in the tides case. It should be negligible but to be concrete I think this would help buttress the conclusions firmly.

"The model was integrated starting from climatological temperature and salinity initial conditions"

It might be worth stating where the climatology was derived from. (is its time mean close to the period of simulation, e.g. how much spin up time does this configuration need are we still looking at largely transient behaviour)

**5 Mixed Layer depth**

The changes if deepened mixed layer depths fit the hypothesis if of the authors in relation to tidal effects. But 13.c also shows shallowing of the mixed layer depth for Tides -No Tides which is difficult to explain in the context enhanced preconditioning.

**7 The Gibraltar Strait**,

I'd be tempted to remove this section as it doesn't really add to the paper. If further analysis it to be added here the authors perhaps could investigate tidal excursions but that is in my option beyond the scope of this particular paper.

**8. Conclusions**

Given the quite short simulation length, longer simulations are really needed to understand the impacts with regards to climate scales and the changes in water mass formation. Furthermore, the feedback with coupling to the atmosphere could be a very interesting next line of inquiry to include in a tides versus no tides comparison. (e.g., what is the effect of different Mixed layer depths etc.)

---

## Author Response (AR1)

**Response to reviewer 1**

We appreciate the time and effort taken to write this review, which has provided us with very useful feedback for this paper. Here we have considered and addressed all comments and questions from the reviewer and believe that we have improved the manuscript overall. Below is the entire reviewer response written in *blue italics* and our changes to the manuscript text in response to the reviewer are highlighted in red.

*The manuscript "The characteristics of tides and their effects on the general circulation of the Mediterranean Sea" by McDonagh et al. presents a study investigating the effect of tides on the Mediterranean Sea circulation. To my knowledge, this subject has not yet been extensively investigated. Apart from Sannino et al. 2014, which provided a first analysis of the tidal influence on the large-scale Mediterranean circulation, previous studies have only focused on specific areas such as the Alboran Sea (Sanchez-Garrido et al., 2013) or the Sicily Strait (Oddo et al., 2023; Gasparini et al., 2004). As such, the present manuscript proposes a valuable contribution to reinforce and deepen the current knowledge on the tidal influence on the Mediterranean Sea.*

*In this study, the authors diagnose the effect of tides from a pair of 5-year-long tidal and non-tidal experiments based on very similar numerical configurations. The first and second sections of the manuscript investigate the influence of tides on the Mediterranean Sea dynamics through the prism of sea level and kinetic energy spectra. These analyses focus on three specific locations: the Strait of Gibraltar, the Tyrrhenian Sea, and the Cretan Sea. They reveal that tides impact high-frequency (> 1 day−1) dynamics through the propagation of the main tidal harmonics at work within the Mediterranean Basin and their interaction with various basin-scale modes. Then, the authors relate the tidally enhanced dynamics and mixing to the deepening of the mixed layer depth in the tidal simulation. The two final sections discuss the impact of tides on the thermohaline properties of the Mediterranean Sea and the transports through the Strait of Gibraltar.*

*Overall, the manuscript has the potential to provide valuable results on tidal contribution to Mediterranean dynamics. However, further work must be done before it can be accepted. More specifically, I think that although the two first sections of the manuscript provide valuable results, they focus on too specific areas to provide an overall picture of the tidal influence on the high-frequency Mediterranean dynamics.*

We have expanded on the earlier sections of the results by providing maps (figure 4) of the sea surface height power spectra at specific frequencies, confirming the regions where tides interact with other phenomena.

*In addition, to assess the influence of tides on the "general circulation of the Mediterranean Sea", as stated in the the title of the study, the manuscript should investigate the influence of tides on the long-term, large-scale circulation.*

The general circulation of the Mediterranean Sea includes both long-term, large-scale circulation, and shorter-term, smaller-scale dynamics. We know already from e.g. Munk and

Wunsch (1998) that small-scale tidal mixing affects the overturning circulation of the ocean, and so the general circulation of the Mediterranean Sea includes both smaller and larger scales. Several decades of model integration, as in Harzallah et al., (2014), would be necessary to begin to understand the effects of tides on the longer-term overturning circulation directly, which was outside of the scope of this study. However, we can infer some potential effects on the large-scale circulation even with a shorter integration.

*Regarding the impact of tidal dynamics, section 5 should more clearly distinguish the influence of local tidal mixing at the strait of Gibraltar, which impacts the Mediterranean mixed layer depth indirectly by changing the density of Atlantic water masses, and the less intense mixing induced by tidal currents throughout the Mediterranean Sea. The former is not directly related to the interaction of tides with the Mediterranean circulation, and it has already been investigated with similar model configurations (Sannino et al., 2014; Naranjo et al., 2014). Thus, I suggest not including it in this manuscript.*

The introduction of local tidal mixing in the Gibraltar Strait is a valuable part of the investigation, because it demonstrates that local tidal mixing has an impact on the large-scale circulation of the whole Mediterranean Sea. The aforementioned literature describes the impacts in and close to the Gibraltar Strait, while our manuscript expands this to the entire Mediterranean Sea.

*On the other hand, the mixing induced by tidal dynamics is relevant to this study. Finally, sections 6 and 7 are, in my opinion, outside of the scope of this paper. Although interesting, the corpus of these sections has no apparent link with Mediterranean circulation and mainly emphasizes the conclusions of previous studies without additional results.*

We have moved section 7 of the manuscript to the supplementary material, while leaving section 6 in the main manuscript (more detail below).

*For these reasons, I am arguing for a major revision of the manuscript. Specifically, I would suggest:*

- *In sections 3 and 4: Add 2D maps of the tidal influence over the specific frequency bands mentioned in the text to give further confidence in the spatial extension of the results discussed.*

We added maps of energy density of sea surface height at 8.2h and 6h. These frequencies were selected as they had large changes in the tidal experiment for the whole Mediterranean, as seen in Figure 2. We added this commentary to go with the new figure (line 152): "Maps of the sea surface height power spectrum at 8.2h and 6h are shown for both experiments in Fig. 4. These frequencies are presented here as they show the largest differences between the two experiments, other than the tidal frequencies directly. Most of the Mediterranean Sea has enhanced power at 8.2h in the tidal experiment (Fig. 4a, c), with particularly large changes in the central Mediterranean: the Tyrrhenian Sea, the Gulf of Gabes, and the Adriatic Sea. The barotropic oscillation at 8.2h is in the Gulf of Gabes, according to Lozano and Candela (1995), but the third mode of the Mediterranean Sea (Schwab and Rao, 1983), and the nonlinear effects of tides in the central Mediterranean Sea according to Palma et al. (2020) are at frequencies close to this (8.4h and 8.0h respectively), and may also have an impact on Figure 4c. Since many of the calculations of barotropic oscillations in the Mediterranean Sea were made several

decades ago, there is a need for an updated confirmation of the frequencies of barotropic oscillations using state-of-the-art methods. In Fig. 4b and 4d, we see that the Sicily escarpment region of Palma et al. (2020) again has particularly enhanced power in the experiment with tides, but other regions such as the Alboran Sea, and the western Mediterranean Sea see an interaction between tides and potential energy at the 6h frequency."

- *In section 5: If you intend to show that the tide-enhanced high-frequency dynamics are responsible for the deepening of the Mediterranean mixed layer depth, you should:*
    - *(1) Mostly focus on the overall increase of the mixed layer depth rather than its local increase over deep convection areas, where the intensity of tidal mixing is unlikely strong enough to drive the deepening.*

We separately show both the mostly increased MLD and mostly increased water mass formation rates in the Mediterranean Sea. The two are related but they are not the same: the deepening of the MLD can be a precursor or element of deep water formation, but it's not the only or most important process. More important precursors include weak stratification, and events such as localised cyclones. We find it interesting that the MLD increase is greatest in the Western Mediterranean Deep Water formation region, in winter, and especially in the years where a lot of deep water is formed, but this isn't enough to establish a connection. We have clarified the separation between these two results in the text (line 223): "The deepening of the mixed layer can be a precursor for increased dense water formation, but a direct analysis of this would be needed to confirm whether tides are enhancing water mass formation as well as mixed layer depth, and we note that the analysis in this work does not directly establish a connection between the two processes. Other important processes such as weak stratification and localised currents are key precursors to deep water formation in the Western Mediterranean."

- - *(2) Look at the seasonal cycle of the mixed layer depth and stratification, as it would be easier to separate the effect of local vertical mixing from that of the tide-induced densification of Atlantic water masses at the Strait of Gibraltar.*

We added a second set of panels to Figure 13 (now Figure 10), showing a timeseries of the mean mixed layer depth and the difference between the two experiments. We also added some commentary for these parts of the figure (line 220): "Regarding the seasonality of mixed layer depth, the absolute and percentage change due to tides is positive throughout the year, but is greatest in winter. We also note that the largest increase in the mean mixed layer depth in the tidal experiment is evident during winter 2019, which is also the year characterised by the largest WMDW formation (see Fig. 11)"

- *Add some results on the influence of tides on the Mediterranean large-scale circulation, or reformulate the title of the article only to consider the high-frequency dynamics.*

As mentioned above, adding results regarding the effects of tides on the long-term circulation is outside the scope of this study, as a much longer model integration would be required. We focus our attention mainly on the smaller spatial and temporal scales in this work, which in turn make up part of the large-scale circulation of the basin.

- *Put sections 6 and 7 in the supplementary materials or a "model validation" section, demonstrating the consistency of the model with previous studies.*

We have moved section 7 to the supplementary material, but have kept section 6 in the main manuscript. We believe this section expands on the work of e.g. Naranjo et al., 2014, which confirmed the higher salinity and lower temperature of the inflowing water from the Gibraltar Strait. Further to this, we find that the changes in the salinity and temperature are consistent in the upper layer of the entire Mediterranean Sea.

- *I do encourage authors to make the necessary effort to improve this manuscript. You can find general and detailed remarks below.*

*General remarks:*

- *There is no model validation. At least a reference to the model validation should be included.*

We have added references to the relevant model validation at the beginning of the Data and Methods section (line 79):

"The general circulation model used is NEMO v3.6, following the implementation of the Mediterranean Sea forecasting system operational in the framework of the Copernicus Marine Service (Coppini et al., 2023; Clementi et al., 2021). The area covered by the model is shown in Fig. 1. The model without tidal forcing is validated in Coppini et al. (2023) and as part of the model information in Clementi et al. (2019), while the experiment with tides is validated in the Quality Information Document in Clementi et al. (2021)."

- *The tidal and non-tidal simulations differ by other processes than tides. Please provide some information about the impact of these differences. The best would be to briefly analyze these impacts in the supplementary materials.*

We added a confirmation in the text that the impact of these differences is negligible compared to the impact of tides (line 121): "The effects of these changes are negligible compared to the effects of adding tidal forcing to the model (see supplementary materials)" This was shown through a preliminary analysis comparing the non-tidal experiment to an experiment, also without tides, which used the timestep and time-stepping scheme of the tidal experiment. We added two figures from this analysis to the supplementary materials, which demonstrate that the effect of changing the time integration scheme is negligible compared to the effect of adding tidal forcing.

- *Text clarity: Please use as few indirect forms as possible to make the manuscript easier to read.*

We have reviewed the writing style of the manuscript to improve the clarity of the text.

*Introduction:*

*General remarks:*
*The introduction is relatively well documented. I thank the authors for this time-consuming work. However, it would benefit from a more straightforward structure. As it is now, the introduction paragraphs discuss:*

1. *Tides in numerical models and their relevance to large-scale circulation*

*I suggest you start with a general introduction, including the content in paragraphs 1 and 4, to introduce the various aspects of tidal waves, their influence on large-scale circulation and mixing, and how they are represented. Then, explain how the tides influence the Mediterranean Sea, as in paragraphs 2, 3, and 5.*

We changed the order of the paragraphs as suggested, beginning with the general content on tides and internal tides in models of the ocean, and then later detailing the influence of tides on specific phenomena and regions within the Mediterranean Sea.

*Also, you should further motivate the need for a deeper understanding of the effect of tidal motion on the Mediterranean circulation. In the current version of the manuscript, this is only mentioned in one sentence "Many of these free oscillations could be affected or enhanced by tides, especially considering their proximity to tidal frequencies."*

We added further motivation to the end of this paragraph (line 44): "Palma et al. (2020) found additionally that spectra of kinetic energy in the Sicily Channel are enhanced at 8 hours and 6 hours due to the nonlinear effects of tides. These works demonstrate the potential importance of interactions between tides and higher frequency features of the Mediterranean Sea, but the interaction between tides and barotropic oscillations in the Mediterranean Sea have not been investigated using a state-of-the-art numerical model."

*Detailed remarks:*
- *"Tidal forcing is a rather recent addition to large scale circulation models that start to have horizontal and vertical resolutions that allow for an analysis of tidal motion on the circulation" => This sentence is unclear. Do you mean that horizontal and vertical resolutions are now fine enough to represent tides properly? Please clarify this.*

We added clarification to this sentence (line 11): "Tidal forcing is a rather recent addition to large-scale circulation models of the ocean, since horizontal and vertical resolutions have become fine enough to allow for an explicit and more accurate representation of tides. This has given rise to novel opportunities to analyse tides and their impacts on the ocean circulation".

- *"Tides are now considered to be essential components of the large scale circulation" => You should provide references specifically investigating the influence of tides on the large-scale circulation, for example: Müller et al., 2010*

We added references (line 14): "Tides are now considered to be essential components of the large scale circulation (St. Laurent et al., 2002; Müller et al., 2010; Melet et al., 2016)"

- *"Recently, Gonzalez (2023) has revisited the tidal dynamics in the Gibraltar Strait, concluding that there are several tidal-induced hydraulic control points and the authors developed a specific mixing parametrization for the Strait." => The Ph.D. of Gonzalez*

*(2023) does not directly investigate hydraulic control points at the Strait of Gibraltar. These were observed by (Farmer & Armi, 1985; Farmer et al., 1988) and discussed by Brandt et al., 1996; Vázquez et al., 2006; Vlasenko et al., 2009; Sánchez-Román et al., 2012; García Lafuente et al., 2013; Hilt et al., 2020.*

We modified the references and reworked this paragraph (line 26): "For the Mediterranean Sea, several authors have depicted the importance of tidal motion for the Gibraltar Strait (Armi and Farmer, 1985; Candela et al., 1990; Harzallah et al., 2014). Armi and Farmer (1985) and Farmer et al. (1988) first observed the hydraulic control points that are induced by tides, and the importance of tidal dynamics and their variability in the region have been more recently discussed by Vázquez et al. (2006), Sánchez-Román et al. (2012), García Lafuente et al. (2013), and Hilt et al. (2020)."

- *"Harzallah et al. (2016) and Naranjo et al. (2014) found that tides at the Strait of Gibraltar: (1) increase the baroclinic volume transport, (2) increase the salinity of Atlantic inflowing waters through the enhancement of mixing, affecting the water mass formation processes further downstream from the Strait and (3) change the Mediterranean deep water outflow." =>*
  - *(1) As I understand it, Harzallah et al. (2016) do not state that tides increase the baroclinic volume transport. In fact, we can read from the abstract of the paper: "It is shown that tidal oscillations reduce the two-way exchange by interaction with the subinertial variability.". The fact that tides increase the baroclinic volume transport is not so evident to me. The paper of Gonzalez et al. 2023 shows that the computed transports through the Strait depend highly on the chosen definition for the interface between inflowing and outflowing waters. Instead of stating that tides increase the baroclinic transports, I suggest that you say they intensify the high-frequency dynamics of the Strait.*

We changed this part to (line 30): "(1) intensify the high frequency dynamics in the Gibraltar Strait"

  - *(2)Tides also cool the Atlantic water masses, although to a lesser extent than they salten it.*

We added this to the sentence (line 31): "2) increase the salinity and, to a lesser extent, decrease the temperature of Atlantic inflowing waters"

  - *(3) You should specify what properties of the Mediterranean deep outflow are changed.*

We added more detail and additional references regarding the outflowing Mediterranean water and how it is affected by tides (line 33): "Tides also change the Mediterranean water outflow, as demonstrated by Izquierdo and Mikolajewicz (2019), where outflowing water moves along the African coast in a model without tides, but currents are closer to climatology with tides, demonstrating a role of tides in the spreading of outflowing water. Moreover, Ambar and Howe (1979) found that tides increase the variability of outflowing salinity"

- *I cannot find van Haren et al. (2014) in the references. Please add it.*

We corrected this error.

- *I think there is a mistake in the reference of the paper of Harzallah et al. (2016), which was published in 2014.*

We corrected this error.

***Data and methods***:

***Detailed remarks***:

- *"Lateral open boundary conditions are used in the Atlantic Ocean and Dardanelles Strait (see Fig. 1)." => What dataset do you use to force the model at these boundaries? Please provide a reference.*

We added further detail and a reference for the boundary conditions of the model (line 87): "Lateral open boundary conditions are used in the Atlantic Ocean and Dardanelles Strait (see Fig. 1), are provided by the Copernicus Marine global analysis and forecast system (Galloudec et al., 2022) for the Atlantic Ocean and a mixture of the aforementioned global model and daily climatology derived from a Marmara Sea box model (Maderich et al., 2015) at the Dardanelles Strait boundary. Further details of the boundary conditions are found in Clementi et al. (2021)."

- *"Additionally, 70 monthly mean climatological freshwater inputs from 39 rivers are added to the surface layer." => Please provide a reference for this dataset.*

We added further detail and a reference for the river climatology dataset (line 90): "Additionally, monthly mean climatological freshwater inputs from 39 rivers are added to the surface layer. Several datasets are used for this: the Global Runoff Data Centre dataset (Fakete et al., 1999) for the Po, Ebro, Nile, and Rhone rivers, the dataset from Raicich (1996) for the Vjosë and Seman rivers, the UNEP-MAP dataset (Demiraj et al., 1996) for the Buna and  rivers, and the PERSEUS dataset (Deliverable of Perseus, 2012) for the remaining 32 Mediterranean rivers which have a mean run-off larger than 50 m3s−1."

- *Please explain how you choose the constant values used in the vertical mixing parameterization.*

We added a reference for this(line 103): "In the model runs used, a and n are 5 and 2 respectively, following Tonani et al (2008)"

***Sea level energy spectra***:

***General remarks***:

*In this section, you should emphasize the innovative aspect of your work more clearly. Oneway to do so would be to discuss separately the interaction of tides with basin-scale frequencies highlighted in previous studies, which you confirm here, and those your study is the first to highlight. Also, a short recap on your findings and their impact at the end of the section would be welcome.*

The extra figure showing 2D maps at specific frequencies adds to the novelty of this section of the work: showing for the first time using state-of-the-art numerical models, maps of the

barotropic oscillations and how they interact with tides. Further to this, we added a recap of the findings as suggested (line 161): "To summarise the analysis of sea surface height, we find that (1) tides affect the sea surface height on spatial and temporal scales away from those of the tides directly, (2) existing Mediterranean basin and regional barotropic oscillations at frequencies with a shorter time period than 12h are excited by tides, particularly at 6h and at several frequencies close to 8h, and (3) maps of the energy density at these frequencies reveal the distribution of these interactions in the Mediterranean Sea."

*Detailed remarks*:
- *"energy is reduced in the tidal model at frequencies lower than 0.5 d−1 (longer than a period of two days)." => Please specify the frequency in days first, as done in the following.*

This has been changed (line 133): "energy is reduced in the tidal model at frequencies with a period longer than two days"

*Kinetic energy spectra*:

*General remarks*:
- *In my opinion, you should remove Figures 7-9 from the corpus of the manuscript. They are used only in a small paragraph, do not provide new information with respect to Figure 6, and focus on a relatively short period of 1 month.*

Most of the information from Figures 7-9 can be seen in either Figure 6, or Figures 10-12. We therefore removed Figures 7-9 and moved some of the discussion of the physical meaning of these plots into the other figures.
- *Figures 10-12 should be replaced by the equivalent of Figure 6 for the vertical velocities. This would make it easier to read the impact of tides on the frequency band mentioned in the text and make the results more robust, as they would integrate the 5 years of simulation instead of one month.*

We decided to keep these figures in their current form. We feel that Figure 12 (now Figure 9) in particular clearly shows the interactions between internal tides and near-inertial waves, where the patterns in the Hovmoller diagram show the wave-wave interactions between them. The rotary spectra in Figure 6 show the frequencies of both types of wave for the kinetic energy. We also note that we do not have hourly data available for the 3-dimensional variables for the entire five-year period, due to computational limitations.

*Detailed remarks*:
- *"We first calculate the rotary spectra for depth-averaged (barotropic) horizontal velocities" => Please include some explanation about the physical meaning of this spectra.*

We added an extra sentence to explain the meaning of the spectra (line 171): "We first calculate the rotary spectra for depth-averaged (barotropic) horizontal velocities (Fig. 5), for both clockwise and counter-clockwise components and then combine these to create the rotary

kinetic energy density spectra. These rotary spectra visualise the time scales at which there is high kinetic energy at each selected point, over the entire water column"

- *"The spectra of kinetic energy density were split into vertical levels to consider baroclinic currents and internal wave modes. [...] characterising the Intermediate Water circulation in the basin." => It is unclear which Figure you refer to. I assume it is Figure 6, but it is confusing since you introduce it later. If the figure you refer to is not in the manuscript, specify it with the mention "(not shown)".*

This does refer to Figure 6. We added clarification (line 177): "We also analysed the kinetic energy spectra split into vertical levels, to consider baroclinic currents and internal wave modes, as shown in Fig. 6."

- *"implying the existence of internal tides as already shown by Gonzalez (2023)." => Here, you should refer to the paper of Gonzalez et al., 2023. Also, many studies investigated internal tides in the Strait of Gibraltar. It is consistent to cite Gonzalez et al. 2023 here, but if you want to cite it alone, you should write something like: "implying the existence of internal tides, recently highlighted in Gonzalez et al. (2023)."*

We added an additional reference here, to an earlier paper on internal tides in the Gibraltar Strait (line 184): "implying the existence of internal tides in the Gibraltar Strait, as shown by Morozov et al. (2002), and more recently by Gonzalez et al. (2023)"

- *"Two zero crossings appear, one approximately at 150m, the lower limit of the inflowing branch of the zonal [...]" => Please detail the implications of this result.*

We added some further information about the importance of this result (line 198): "Two zero crossings appear, one approximately at 150m, the lower limit of the inflowing branch of the zonal conveyor belt already described above (Pinardi et al., 2019), and the second at 300m. This is particularly apparent in the Gibraltar Strait (Fig. 10), where internal tide generation leads to increased tidal velocity in both directions. The continuation of this zero-crossing at 150m in other regions (Figs. 8-9) demonstrates the importance of internal tide generation at the Gibraltar Strait and how it affects the entire Mediterranean Sea general circulation, including in remote regions."

***Mixed layer depth and water mass formation***:

***Detailed remarks***:
*"There is an increase in mixed layer depth with tides throughout most of the basin, with notable exceptions in the Gibraltar Strait/Alboran Sea region, and in parts of the Aegean Sea." => Please explain why the mixed layer depth responds differently to tides in these regions.*

We added further context for this result (line 213): "There is an increase in winter mixed layer depth with tides throughout most of the basin, with notable exceptions in the Gibraltar Strait/Alboran Sea region, and in parts of the Aegean Sea. In the Gibraltar Strait, this is explained by a thicker interface layer (Garcia-Lafuente et al., 2013), which shrinks the upper layer compared to the experiment without tides, reducing the depth at which the threshold density change for the mixed layer is reached. These effects extend into the upper-layer Alboran Sea

water which originates in the Gibraltar Strait. In the Aegean Sea, despite having low amplitude barotropic tides, internal tides are present (Alford et al., 2012), and affect mixing at the bottom (Gregg et al., 2012) which likely in turn affects the water column structure in this shallow region."

*Temperature and salinity:*

*Detailed remarks:*
*"As indicated in work by Naranjo et al. (2014) and Harzallah et al. (2016), inflowing salinity at Gibraltar increases when tides are introduced and upper layer temperature decreases" => You should also cite the papers of Gonzalez et al 2023, Sanchez-Roman et al., 2018, and Sannino et al 2014,*
We added the additional references suggested (line 328): "As indicated in work by Naranjo et al. (2014), Harzallah et al. (2014), Sanchez-Roman et al. (2018), and Gonzalez et al. (2023), inflowing salinity at Gibraltar increases…"

*The Gibraltar Strait:*

*General remarks:*
*You should specify how you compute the inflow and outflow transports through the strait of Gibraltar. As discussed in Gonzalez et al. 2023, it significantly impacts the transport values obtained.*
We detailed the method used to calculate these values (supplementary material): "Net water mass transport through the strait at a given longitude, over the five-year period, was calculated as … where Q is the net water mass transport, η is the sea surface height, z is the depth, and u is the mean zonal velocity. This transport can be split into eastward and westward components: … where Qin is the upper-layer eastward transport, Qout is the lower-layer westward transport, and H is the Heaviside step function, H = 1 if u > 0, H(u) = 0 otherwise". The ellipses here are equations which are added to the supplementary material of the manuscript, along with this entire section.

*Detailed remarks:*
*"Gonzalez (2023) recently demonstrated that the required resolution for an accurate representation of the Gibraltar Strait would be about five times the one used in our model." => It is Sannino et al., 2009 who discuss the resolution needed to represent the Strait of Gibraltar, not Gonzalez et al., 2023.*
We corrected this reference (supplementary material): "We argue that this is due to the relatively coarse resolution of the model in the Gibraltar Strait, since Sannino et al. (2009) demonstrated that the required resolution for an accurate representation of the Gibraltar Strait would be at the sub-kilometer scale, many times greater than the one used in our model"

***Figures***:

*Fig1: Please adapt the colormap so it is possible to see the tidal amplitude in the Atlantic. I understand tidal amplitudes significantly differ over the Mediterranean and the Atlantic, making it difficult to plot both with one colormap. However, you could use one colormap for the Atlantic and one for the Mediterranean. Also, add the tidal phase in contours to display amphidromic points.*

We updated this figure to include the phase and colourmap in the Atlantic Ocean.

*Fig. 2: Add the scale in days for the two lower panels. Please indicate the frequency you refer to in the text using dashed lines and display the labels of tidal harmonics next to the associated peaks.*

We updated the figure with the additional barotropic oscillations and tidal harmonics.

*Fig. 3: Please indicate the frequency you refer to in the text using dashed lines and display the labels of tidal harmonics next to the associated peaks.*

We updated the figure.

*Fig. 4: I do not find any reference to the bathymetry in the manuscript, so you should indicate the points specified here in Figure 1 and remove this figure.*

We removed this figure and added the points to Figure 1.

*Fig. 6: Please add a panel displaying the differences between the two simulations.*

We added this panel.

**Response to reviewer 2**

We appreciate the efforts made for this very useful review of our manuscript. Here we have addressed each comment made and detailed the changes to the manuscript. Below is the entire reviewer response written in *blue italics* and our changes to the manuscript text in response to the reviewer are highlighted in red.

*Overview of the paper*:
*This paper undertakes an assessment on a basin scale of the effects of tides on the Mediterranean Sea's circulation. The methodology involves an analysis of two experiments, one with tides and one without. It should be noted that tides aside, the experiments do have some other differences. To my knowledge there has not been such a systematic analysis of the effect of tides to the Mediterranean basin as a whole and expands on several studies which focus on specific sub regions. Thus, the subject matter of this paper is both very relevant to the journal and makes a substantial contribution beyond what has been done to date.*
*The twin experiments are of 5 years duration. To understand the effect on the Mediterranean dynamics the first parts of the paper focus on sea level and kinetic spectra in the following locations, the Strait of Gibraltar, the Tyrrhenian Sea, and the Cretan Sea. Clear Amplification of several sub daily modes is demonstrated by the analysis. The next section goes on to investigate how this tidal amplification leads to deepening of the mixed layer depth. To finish the paper discusses the effects of the tides on the thermohaline structure of the Mediterranean and briefly the baroclinic transport through the Strait of Gibraltar.*

*General constructive critiques*:
*The sections on the energy spectra are in the end limited to specific locations of interest. I can see why this is done as a pragmatic decision, but it lessens the impact of the paper which is aimed at a more general analysis of impact to the basin scale circulation. Perhaps a broader scale analysis in complement to the site-specific analysis could be done to shore up the results here and conclusions inferred from them such as broader scale maps?*
It's not possible to take the mean of spectra over the whole Mediterranean basin or large regions when studying internal waves, because internal waves are generated according to the topography and can have a different vertical structure at each gridpoint. Therefore, averaging over many gridpoints could remove the signature of the internal waves from the spectra entirely. To avoid this, we decided to select individual gridpoints for this analysis. However, we feel that our new addition of maps of potential energy from sea surface height at specific frequencies (figure 4) further enhances the basin-scale analysis of the paper and complements the analysis already shown in the initial submission.

*It might be worth raising limitations of the model at only 4km resolution with a hydrostatic model to faithfully represent internal tides. It wouldn't detract from the paper overall as the point is increased tidal energetics exciting existing modes. But of course, the accuracy of the internal*

*tides themselves would be questionable in such a configuration and miss some key processes. The point here is to bring such limitations of the configuration to the attention of the reader so they do not over infer the models representation of internal tides.*

We added a short paragraph to the conclusion discussing this limitation of the model with regards to its ability to resolve internal tides (line 276): "It is important to note that the resolution of this model configuration, although higher than many global ocean models and regional ocean models from past studies, has limitations in its representation of waves with shorter wavelengths, including internal tides. According to Li et al. (2015, 2017), the first two modes of the M2 internal tide and first three modes of the K1 internal tide all have wavelengths greater than 45 km, but higher modes are likely unresolved at the 1/24° resolution. Furthermore, high vertical resolution is important for simulating internal tides. Our model lacks high vertical resolution at deeper levels, and this could contribute to a lack of internal tides in the deeper ocean."

*The twin experiments do have some differences other than the tides. I understand that the differences will probably be very small compared to the tides themselves, but I think it is worth at least conducing as short simulation (order 1 month) of the no tides solution with the exact same options as the tide simulation to confirm that the other differences are negligible.*

We added a confirmation in the text that the impact of these differences is negligible compared to the impact of tides (line 121): "The effects of these changes are negligible compared to the effects of adding tidal forcing to the model (see supplementary materials)" This was shown through a preliminary analysis comparing the non-tidal experiment to an experiment, also without tides, which used the timestep and time-stepping scheme of the tidal experiment. We added two figures from this analysis to the supplementary materials, which demonstrate that the effect of changing the time integration scheme is negligible compared to the effect of adding tidal forcing.

*The final 2 sections are quite abrupt and may in the end not particularly conclusive in either case. That said "negative results" are also of value and the suggestion with regards to resolution for 7 is a very reasonable further line of inquiry. I would not demand that the sections are removed I just feel they may not in their current form be adding anything substantial to the paper in its current form and may be better left in 8 as future lines of inquiry to investigate further.*

We have moved section 7 to the supplementary material, but have kept section 6 in the main manuscript. We believe this section expands on the work of e.g. Naranjo et al., 2014, which confirmed the higher salinity and lower temperature of the inflowing water from the Gibraltar Strait. Further to this, we find that the changes in the salinity and temperature are consistent in the upper layer of the entire Mediterranean Sea.

**Section specific points**
*(general tiny niggle, this is picky and trivial, but can the units and numbers be spaced out e.g. 3.5 TW instead of 3.5TW etc.)*

We have corrected this in the text

**1 Introduction**

*The introduction in my view does a good job of introducing the relevant background research. It might be worth mentioning the increasing importance of tides in coupled climate simulations, though the Arbic reference also goes into this, and perhaps that this is important for future simulations of the Mediterranean particularly at climate time scales if the water formation regions are affected.*

We added a sentence to the end of the first paragraph of the introduction (line 16) to mention this issue: "Tides are also an important phenomenon in coupled numerical models, which represents a new opportunity in high-resolution modelling of the ocean and atmosphere (Arbic, 2022), and long-term simulations"

**2 Data and Methods**

*As mentioned above it might be nice to rule out any differences induced by the time stepping by a short run of the no tides simulation with the exact settings and bathy as used in the tides case. It should be negligible but to be concrete I think this would help buttress the conclusions firmly.*

We added a confirmation in the text that the impact of these differences is negligible compared to the impact of tides (line 121): "The effects of these changes are negligible compared to the effects of adding tidal forcing to the model (see supplementary materials)". This was shown through a preliminary analysis comparing the non-tidal experiment to an experiment, also without tides, which used the timestep and time-stepping scheme of the tidal experiment. We added two figures from this analysis to the supplementary materials, which demonstrate that the effect of changing the time integration scheme is negligible compared to the effect of adding tidal forcing.

*"The model was integrated starting from climatological temperature and salinity initial conditions" It might be worth stating where the climatology was derived from. (is its time mean close to the period of simulation, e.g. how much spin up time does this configuration need are we still looking at largely transient behaviour)*

We added a reference for the climatology used for the initial conditions (line 127): "The model was integrated for seven years (2015-2021), starting from climatological temperature and salinity initial conditions derived from the winter climatology (2005-2012) of the World Ocean Atlas (Boyer et al., 2013). The first two years are removed from the following analysis as they are considered as a spin-up period."

**5 Mixed Layer depth**

*The changes if deepened mixed layer depths fit the hypothesis if of the authors in relation to tidal effects. But 13.c also shows shallowing of the mixed layer depth for Tides -No Tides which is difficult to explain in the context enhanced preconditioning.*

We added further context for this result (line 213): "There is an increase in winter mixed layer depth with tides throughout most of the basin, with notable exceptions in the Gibraltar Strait/Alboran Sea region, and in parts of the Aegean Sea. In the Gibraltar Strait, this is explained by a thicker interface layer (Garcia-Lafuente et al., 2013), which shrinks the upper layer

compared to the experiment without tides, reducing the depth at which the threshold density change for the mixed layer is reached. These effects extend into the upper-layer Alboran Sea water which originates in the Gibraltar Strait. In the Aegean Sea, despite having low amplitude barotropic tides, internal tides are present (Alford et al., 2012), and affect mixing at the bottom (Gregg et al., 2012) which likely in turn affects the water column structure in this shallow region."

We also added extra panels to this figure to show the seasonality of mixed layer depth in the whole Mediterranean Sea.

*7 The Gibraltar Strait,*

*I'd be tempted to remove this section as it doesn't really add to the paper. If further analysis it to be added here the authors perhaps could investigate tidal excursions but that is in my option beyond the scope of this particular paper.*

We have moved section 7 to the supplementary material, and added further detail about the method used to calculate the transport.

*8. Conclusions*

*Given the quite short simulation length, longer simulations are really needed to understand the impacts with regards to climate scales and the changes in water mass formation. Furthermore, the feedback with coupling to the atmosphere could be a very interesting next line of inquiry to include in a tides versus no tides comparison. (e.g., what is the effect of different Mixed layer depths etc.)*

We added a sentence to the conclusions regarding the need for longer simulations to understand the overturning circulation and its interaction with tides (line 287): "In addition, a deeper focus on the impacts of tides, both barotropic and baroclinic, on vertical motion in the most vertically dynamic regions of the Mediterranean Sea would further help in understanding the overturning circulation in the basin. This could include considering the impact of tides on vertical mixing in deep and intermediate water formation regions, and how parameterization choices in numerical models may affect this. However, a longer simulation of several decades would be required to reveal the impact of tides on the overturning circulation of the Mediterranean Sea."

---

## Referee Report (RR1)

**Second review of "The characteristics of tides and their effects on the general circulation of the Mediterranean Sea" by McDonagh et al.**

**Context**

The manuscript "The characteristics of tides and their effects on the general circulation of the Mediterranean Sea" by McDonagh et al. presents a study investigating the effect of tides on the Mediterranean Sea circulation. To my knowledge, this subject has not yet been extensively investigated. Apart from Sannino et al. 2014, which provided a first analysis of the tidal influence on the large-scale Mediterranean circulation, previous studies have only focused on specific areas such as the Alboran Sea (Sanchez-Garrido et al., 2013) or the Sicily Strait (Oddo et al., 2023; Gasparini et al., 2004). As such, the present manuscript proposes a valuable contribution to reinforce and deepen the current knowledge on the tidal influence on the Mediterranean Sea.

In this study, the authors diagnose the effect of tides from a pair of 5-year-long tidal and non-tidal experiments based on very similar numerical configurations. The first and second sections of the manuscript investigate the influence of tides on the Mediterranean Sea dynamics through the prism of sea level and kinetic energy spectra. The first section first investigates the whole Mediterranean Sea and then focuses on the Adriatic Sea and the Gulf of Gabes. The second section focuses on three specific locations: the Strait of Gibraltar, the Tyrrhenian Sea, and the Cretan Sea. They reveal that tides impact high-frequency (> 1 day$^{-1}$) dynamics through the propagation of the main tidal harmonics at work within the Mediterranean Basin and their interaction with various basin-scale modes. Then, the authors relate the tidally enhanced dynamics and mixing to the deepening of the mixed layer depth in the tidal simulation. The final sections discuss the impact of tidal transformation of Atlantic water masses at the Strait of Gibraltar on the thermohaline properties of the Mediterranean Sea.

**Second review: summary**

For the first round of review, I requested additional materials from the authors to strengthen the highlighted results. I am pleased to acknowledge the authors' efforts in modifying the manuscript. Their work has enriched the study with valuable information. However, these additional materials also raise critical questions about the main results, making them more fragile. To be specific :

In section 3, the frequencies mentioned in the text are now presented in Figures 2, 3, and 4, improving the clarity of the analysis. However, this also brings to light several discrepancies between the results highlighted in the text and the content of the figures. There are also

inconsistencies between the investigated simulations and the literature, which is nevertheless used to interpret the physical meaning of the results.

In section 5, the two new panels of Figure 10 show that mixed layer differences between the simulations mainly result from mixed layer deepening in winter. In the literature, tidal deepening of winter mixed layer, associated with deep and intermediate convection, has been discussed in several studies (PhD of Gonzalez et al., 2023; Sannino et al., 2015; Naranjo et al., 2014), in particular over the Gulf of Lion. So far, it has always been related to the densification of Atlantic water masses by tidal mixing at the Strait of Gibraltar. This remote effect of tides is out of the scope of the present study, which investigates the influence of tidal dynamics in the Mediterranean Sea. Thus, to focus on mixed layer depth variations at the Gulf of Lion, the authors must show explicit evidence that the observed signal is at least partly related to local tidal dynamics. In the current version of the manuscript, the authors only specify that mixed layer changes over the Gulf of Lion may not be related to local tidal mixing.

Besides the above points, I still have issues with section 4, in which the authors chose not to include additional diagnostics that reviewer #2 and myself recommended. In the current version of the manuscript, the limitation of the horizontal kinetic energy analysis to three discrete points (in the Strait of Gibraltar, the Cretan Sea, and the Tyrrhenian Sea) makes the results hardly generalizable to the rest of the Mediterranean. The vertical currents analysis is even more problematic to me as it only relies on a one-month temporal window to investigate the impact of tidal dynamics on the sea vertical currents. On such a short temporal window, and focusing only on three discrete points in space, it seems impossible for me to properly disentangle the influence of tide from the internal variability of the ocean circulation, which is not mentioned in the analysis.

For these reasons, I cannot recommend the manuscript for publication in its actual form. The manuscript's body should be comprehensively revised to address the points above. Because the associated modifications will require new sets of diagnostics for sections 4 and 5 and an in-depth modification of section 3, I feel it is better to start a new review process with an updated manuscript. Therefore, I am arguing for a rejection of the current manuscript. I am very sorry for the inconvenience that goes along. Below is a detailed list of the issues I have noted and some recommendations. I sincerely hope the authors can find the time to make the necessary modifications to publish this valuable study.

**Detailed comments**

**General:**

The authors have chosen to keep section 6 in the body of the manuscript, arguing: «This section expands on the work of, e.g., Naranjo et al., 2014, which confirmed the higher salinity and lower temperature of the inflowing water from the Gibraltar Strait. Further to this, we find that the changes in the salinity and temperature are consistent in the upper layer of

the entire Mediterranean Sea. ». I respect their perspective, but I hold a different view on this matter. In my opinion, section 6 should not be included in the body of the manuscript because :

1. This study is meant to investigate the influence of tides on the Mediterranean Sea dynamics and the associated impact on its hydrography. However, this section only discusses the effect of Atlantic water mass transformation at the Strait of Gibraltar, which is unrelated to local tidal modulations of the Mediterranean circulation.

2. The consistency of the temperature and salinity changes over the Mediterranean Sea has already been investigated by Sannino et al., 2015 and Harzalla et al. 2014.

However, I think this section provides valuable information in a « model validation » section or as supplementary material.

**Introduction**

**General remarks:**
The clarity of the introduction has much improved. I thank the authors for their time in making these modifications. Below are some minor remarks, mainly to improve the references.

**Detailed remarks:**
- L. 20: "The presence of internal tides generated in the Gibraltar Strait is discussed in Morozov et al. (2002) and Vlasenko et al. (2009)." => You should add the recent paper of Roustan et al., 2024, and Hilt et al., 2020.
- L.25: "For the Mediterranean Sea, several authors have depicted the importance of tidal motion in the Gibraltar Strait (Armi and Farmer, 1985; Candela et al., 1990; Harzallah et al., 2014)." => You should also cite Sannino et al., 2015 and Naranjo et al., 2014.
- L35: "Moreover, 35 Ambar and Howe (1979) found that tides increase the variability of outfowing salinity." => This reference is pretty old. It would be best to back it up with a more recent study.
- L55: "To our knowledge, the effects of tides on the circulation of the entire Mediterranean Sea has not been extensively investigated." => Sannino et al., 2014 did investigate this (see section 4).

**Data and methods**
**General remarks**:
Thank you for taking the time to modify this section. I have nothing more to say about it.

**Sea level energy spectra**

**General remarks**:

I thank the authors for taking the time to include my suggestions. Because of the issues mentioned above (more details below), I think this section should be entirely rewritten to provide a more consistent physical interpretation of the frequencies enhanced by tides. This may also be an opportunity to give a critical view of the literature, in which several aspects investigated in this study have not been updated in a long time.

**Detailed remarks :**

L140 – 145: Broad 12h energy peak

"We argue that the broad 12h energy peak in Figure 2 in the tidal run is composed of the amplified 11.4h Mediterranean Sea basin mode energy (Schwab and Rao, 1983), the first Adriatic mode at 10.7h, known to be enhanced by tides (Medvedev et al., 2020; Schwab and Rao, 1983), and the 12h Adriatic/Aegean seas mode (Lozano and Candela, 1995)." => Now that the 11.4h and 10.7h frequencies are explicitly shown in Figure 2, we can see that they are on the upper end of the broad 12h peak and, thus, can only explain the upper part of the peak. In addition, the mentioned frequencies are not specifically enhanced in the non-tidal simulation, and no significant peak is visible in the tidal simulation. In my opinion:

- You must revise the 12h broad peak physical interpretation, as it does not appear related to the mentioned frequencies.
- The 11.4h and 10.7h energy modes should not be interpreted as tidally enhanced, as no clear signal is visible in the tidal and non-tidal experiments. Note that no signal is visible in the Adriatic Sea for the 10.7h Adriatic mode (Figure 3. a). In this regard, explaining why the simulations do not represent these basin mode frequencies would be valuable.

 L145: 8h energy peak

"The peak at 8h also aligns with the Western Mediterranean basin mode of 8.4h discussed in Schwab and Rao (1983), and the Gulf of Gabes mode at 8.2h (Lozano and Candela, 1995). Schwab and Rao (1983) also noted a fourth Mediterranean mode at 7.4h and the third Adriatic mode at 6.7h."=> The 8.4h and 8.2h are distinct from the 8h peak, so they should not be used to explain it. Also, in Figure 3b, the 8.2h frequency does not seem specifically intensified over the Gulf of Gabes, so its physical interpretation should be revised (see following comments). The 7.4h and 6.7h peaks are not intensified; if they are mentioned, you should explain why they are not present in the simulations.

L150: Adriatic Sea

"In the Adriatic Sea (Fig. 3a), the sea level energy peaks at the frequencies of the barotropic modes of the Adriatic Sea at 11.4 hours, 6.7 hours (Schwab and Rao, 1983) and 12 hours (Lozano and Candela, 1995) are all enhanced by tides. The peaks are also visible in the model without tides, but with lower energy, thus confirming that these peaks are due to amplification of existing modes forced by winds and atmospheric pressure." => The 6.7h peak is not particularly high in the non-tidal simulation and is not intensified in the tidal simulation. You should display the 11.4h frequency in Figure 3. a. In addition, it does not seem to be significantly intensified, neither in the non-tidal simulation nor in the tidal one.

L150: Gulf of Gabes
"The barotropic oscillation at 8.2h is in the Gulf of Gabes, according to Lozano and Candela (1995), but the third mode of the Mediterranean Sea (Schwab and Rao, 1983), and the nonlinear effects of tides in the central Mediterranean Sea according to Palma et al. (2020) are at frequencies close to this (8.4h and 8.0h respectively), and may also have an impact on Figure 4c. Since many of the calculations of barotropic oscillations in the Mediterranean Sea were made several decades ago, there is a need for an updated confirmation of the frequencies of barotropic oscillations using state-of-the-art methods." => Thanks for this paragraph, which clarifies the discrepancies between the literature and the manuscript. Since you highlight these discrepancies, the 8.2h frequency should not be associated only with the Gulf Gabes in these simulations.

**Kinetic energy spectra**

**General remarks:**
For the reasons mentioned above, I suggest entirely rewriting this section. The updated section could first investigate the kinetic energy at the Mediterranean scale (as in the first section) and then focus on specific frequency bands impacted by tides, providing 2D maps of their influence. Regarding the vertical velocities, I understand that 3D hourly outputs are complex to handle, but one month of data is insufficient to conclude on the effects of tides.

**Mixed layer depth and water mass formation**

**General remarks:**
I am sorry that investigating the mixed layer depth seasonal cycle did not make the separation of local and remote effects of tides easier. I think this section is interesting, but further work is needed, as it is essential to properly isolate the local influence of tidal dynamics on the mixed layer depth. In this regard, another simulation, which "sees" the same Atlantic water masses as the non-tidal one at the entrance of the Mediterranean, but still includes Mediterranean tides, could be used. Diagnosing the impact of the modulated stratification due to the Atlantic water masses transformation at the Strait of Gibraltar on the mixed layer depth variation to deduce the local effect of tides as a residual could be another solution (see Sannino et al., 2009, section 4.3).

**References:**

Brandt P, Alpers W, Backhaus JO (1996) Study of the generation and propagation of internal waves in the Strait of Gibraltar using a numerical model and synthetic aperture radar images of the European ERS 1 satellite. Journal of Geophysical Research: Oceans 101(C6):14,237–14,252, DOI 10.1029/96JC00540, URL http://doi.wiley.com/10.1029/96JC00540

Farmer DM, Armi L (1985) The internal hydraulics of the Strait of Gibraltar and associated sills and narrows. Oceanologica acta 8(1):37–46, URL https://archimer.ifremer.fr/doc/00112/22325/

Farmer DM, Armi L, Armi L, Farmer DM (1988) The flow of Atlantic water through the Strait of Gibraltar. Progress in Oceanography 21(1):1, DOI 10.1016/0079-6611(88)90055-9, URL https://doi.org/10.1016/0079-6611(88)90055-9

Garcia Lafuente J, Bruque Pozas E, S´anchez Garrido JC, Sannino G, Sammartino S (2013) The interface mixing layer and the tidal dynamics at the eastern part of the Strait of Gibraltar. Journal of Marine Systems 117-118:31–42, DOI 10.1016/j.jmarsys.2013.02.014, URL http://dx.doi.org/10.1016/j.jmarsys.2013.02.014

Gasparini GP, Smeed DA, Alderson S, Sparnocchia S, Vetrano A, Mazzola S (2004) Tidal and subtidal currents in the Strait of Sicily. Journal of Geophysical Research: Oceans 109(2), DOI 10.1029/2003jc002011

Gonzalez N, Waldman R, Sannino G, Giordani H, Somot S (2023) Understanding tidal mixing at the Strait of Gibraltar: A high-resolution model approach. Progress in Oceanography 212:102,980, DOI 10.1016/j.pocean.2023.102980, URL https://linkinghub.elsevier.com/retrieve/pii/S007966112300023X

Harzallah A, Alioua M, Li L (2014) Mass exchange at the Strait of Gibraltar in response to tidal and lower frequency forcing as simulated by a Mediterranean Sea model. Tellus A: Dynamic Meteorology and Oceanography 66(1):23,871, DOI 10.3402/tellusa.v66.23871, URL https://doi.org/10.3402/tellusa.v66.23871

Hilt M, Auclair F, Benshila R, Bordois L, Capet X, Debreu L, Dumas F, Jullien S, Lemari´e F, Marchesiello P, Nguyen C, Roblou L (2020) Numerical modelling of hydraulic control, solitary waves and primary instabilities in the Strait of Gibraltar. Ocean Modelling 151(May):101,642, DOI 10.1016/j.ocemod.2020.101642, URL https://linkinghub.elsevier.com/retrieve/pii/S146350032030144X

Muller M, Haak H, Jungclaus JH, S¨undermann J, Thomas M (2010) The effect of ocean tides on a climate model simulation. Ocean Modelling 35(4):304–313, DOI 10.1016/j.ocemod.2010.09.001, URL https://linkinghub.elsevier.com/retrieve/pii/S1463500310001289

Naranjo C, Garcia-Lafuente J, Sannino G, Sanchez-Garrido JC (2014) How much do

tides affect the circulation of the Mediterranean Sea? From local processes in the Strait of Gibraltar to basin-scale effects. Progress in Oceanography 127:108–116, DOI 10.1016/j.pocean.2014.06.005

Roustan, Jean-Baptiste & Bordois, Lucie & Lafuente, J. & Dumas, Franck & Auclair, Francis & Xavier, Carton. (2024). Evidence of Reflected Internal Solitary Waves in the Strait of Gibraltar. Journal of Geophysical Research: Oceans. 129. 10.1029/2023JC020152.

Oddo P, Poulain PM, Falchetti S, Storto A, Zappa G (2023) Internal tides in the central Mediterranean Sea: observational evidence and numerical studies. Ocean Dynamics 73(3-4):145–163, DOI 10.1007/s10236-023-01545-z, URL https://doi.org/10.1007/s10236-023-01545-z https://link.springer.com/10.1007/s10236-023-01545-z

Sanchez-Garrido JC, Garc´ıa Lafuente J, ´Alvarez Fanjul E, Sotillo MG, de los Santos FJ (2013) What does cause the collapse of the western alboran gyre? results of an operational ocean model. Progress in Oceanography 116:142–153, DOI 10.1016/j.pocean.2013.07.002, URL http://dx.doi.org/10.1016/j.pocean.2013.07.002

Sanchez-Román A, García-Lafuente J, Delgado J, S´anchez-Garrido JC, Naranjo C (2012) Spatial and temporal variability of tidal flow in the Strait of Gibraltar. Journal of Marine Systems 98-99:9–17, DOI 10.1016/j.jmarsys.2012.02.011, URL http://dx.doi.org/10.1016/j.jmarsys.2012.02.011

Sanchez-Roman A, Jorda G, Sannino G, Gomis D (2018) Modelling study of trans-formations of the exchange flows along the Strait of Gibraltar. Ocean Science 14(6):1547–1566, DOI 10.5194/os-14-1547-2018

Sannino G, Garrido JC, Liberti L, Pratt L (2014) Exchange Flow through the Strait of Gibraltar as Simulated by a σ-Coordinate Hydrostatic Model and a z-Coordinate Nonhydrostatic Model. The Mediterranean Sea: Temporal Variability and Spatial Patterns 9781118847:25–50, DOI 10.1002/9781118847572.ch3

Vazquez A, Stashchuk N, Vlasenko V, Bruno M, Izquierdo A, Gallacher PC (2006) Evidence of multimodal structure of the baroclinic tide in the Strait of Gibraltar. Geophysical Research Letters 33(17):L17,605, DOI 10.1029/2006GL026806, URL http://doi.wiley.com/10.1029/2006GL026806

Vlasenko V, Sanchez Garrido JC, Stashchuk N, Lafuente JG, Losada M (2009) Three-dimensional evolution of large-amplitude internal waves in the Strait of Gibraltar. Journal of Physical Oceanography 39(9):2230–2246, DOI 10.1175/2009JPO4007.1

---

## Author Response (AR2)

**Response to editor**

Thank you for your review of this work, we appreciate the time and effort put into it. Here we have addressed each of your comments in turn. We have included your entire review in *blue italics*, with our responses in black and specific changes to the text in red.

*Dear authors,*

*thank-you for your patience while I found an opportunity to consider the review of your latest manuscript. As you will have seen, although the review was positive about many aspects of the paper, the overall assessment was that the paper needed major revisions or perhaps such a substantial rewrite that it should be rejected and resubmitted.*

*I do not think this is necessary, and that although there is some further work required for publication it is relatively minor. The reviewer has made several suggestions for further work, but I think that these can be left for the future. It is better to publish the work that has already been done and ensure it is available for others. There will always be more that can be done and more ways to present such a complex data set. Of course if you agree with the reviewer and wish to do more then you may do so, but if you consider the changes unnecessary I am happy to proceed with substantially the same paper (with a few exceptions, please see below).*

*For now I will select "minor revisions", which will trigger a short automated deadline. If you need more time then please just ask the editorial office to push the deadline back as required.*

Thank you for this feedback. We found some of the reviewer's comments very useful and will certainly consider them in future work. We added a paragraph to the end of the conclusion to describe possible future work directly, to acknowledge what was not done in this work, and to motivate future work in this area.

*==========================*
*The following clarifications and changes are necessary:*

*Please ensure that if there are caveats about the strength of the evidence that tides cause the mixed depth changes (eg that are in the main text in Section 5), they are also in the conclusions and abstract. It would help to clarify to have a specific subheading for caveats in the conclusions.* We added a sentence to the conclusion to emphasise this point: "It should be noted that although an increased mixed layer depth is one conditioning element for dense water formation, other concomitant processes are needed for deep water formation events to occur (Marshall and Schott, 1999; Pinardi et al., 2022).", and in the abstract, we changed the last sentence to "Tides increase the mixed layer depth in the Mediterranean Sea, particularly in the deep and intermediate water formation areas of the Western and Eastern basins. The addition of tides, the cases considered, does also enhance Western Mediterranean Deep Water formation."

*Additional references in the introduction are not necessary, there is no need to cite every paper.*
We added some of the citations suggested by the reviewer, such as Hilt et al. (2020).

*Regarding the 12h peak (line 140), I interpret this as being not just a sharp 12h peak but a dicussion of a broad semi-diurnal frequency range that arises from spatial averaging over the basin. Is that correct? If so, please ensure it is clear but I agree with your physical explanation. If not then there is a problem with confusing frequencies.*
There are several explanations for the broad peak, and the true reason is likely a combination of several of the following: firstly the combination of the peaks of the four semidiurnal tidal components in this model, and their blurring due to both spatial averaging and the temporal resolution of the experiment output being only hourly and not more frequent. Secondly, we have the amplification of basin/regional modes at 11.4h and 12h and their interactions with the tides, and thirdly there are further possible interactions of tides with other mesoscale phenomena in the Mediterranean Sea, particularly eddies. We can't confirm the exact interactions with this plot alone, but we begin to discuss these ideas in this paragraph. I think some of the frequencies did get mixed up here when we changed things after the first review. We edited this paragraph, both to address your later comment regarding line 144*, and to also highlight the possibility of interactions between tides and other mesoscale phenomena as a further reason for the broad peak: "In the semidiurnal range, the tidal model introduces a broad peak around 12 hours, as well as peaks at the individual tidal component frequencies. This is due to the amplification by tides of the basin modes near and at these frequencies. We argue that the broad 12h energy peak in Figure 2 in the tidal run is composed of the amplified 11.4h Mediterranean Sea basin mode energy (Schwab and Rao, 1983) and the 12h Adriatic/Aegean seas mode (Lozano and Candela, 1995). Moreover, interactions between internal tides and other mesoscale phenomena such as eddies could be affecting the internal tidal kinetic energy, as discussed in Guo et al. (2023), but a dedicated analysis would be required to confirm whether this is the case in the Mediterranean Sea. Furthermore, the spatial and temporal averaging of model outputs has likely blurred these peaks slightly."

*Fig 2, caption: This must be the spatial average of the point-wise energy spectrum, not the energy spectrum of the spatial average? Be precise.*
We added a sentence to the caption to detail the calculation of the spectrum: "The basin-mean spectrum is calculated as an area-weighted mean of the periodogram at each gridpoint in the Mediterranean Sea."

*Line 144: I agree with the reviewer that the 10.7h signal is missing from the Adriatic in figure 3a, please check.*
* See above

*Line 147: the wording needs to be clearer about the separate 8.4h, 8.2 and 8h peaks.*

We have rewritten this paragraph to improve precision: "Peaks are also seen at 8h and 6h, which correspond to the kinetic energy frequencies in the Sicily Strait, noted by Palma et al. (2020) where these peaks were considered to come from non-linear effects of tides. Further peaks that are enhanced by tides align with the Western Mediterranean basin mode of 8.4h discussed in Schwab and Rao (1983), and the Gulf of Gabes mode at 8.2h (Lozano and Candela, 1995). Schwab and Rao (1983) also noted a fourth Mediterranean mode at 7.4h and the third Adriatic mode at 6.7h, but these are not visible in the basin-mean spectrum of Fig. 2 and are not affected by tides. Overall, there are many oscillations in the Mediterranean Sea that are interacting with tides leading to their enhancement, although not every barotropic oscillation mentioned in the literature is affected by tides."

*Line 152: I agree with the reviewer that the 6.7hour Adriatic mode does not appear to be enhanced by tides, please check.*

We have rewritten this paragraph: "In the Adriatic Sea (Fig. 3a), the sea level energy peaks at the frequencies of the barotropic modes of the Adriatic Sea at 10.7 hours and 12 hours (Lozano and Candela, 1995) are enhanced by tides, whereas the mode at 6.7 hours (Schwab and Rao, 1983) has a similar energy density in both experiments. This is evidenced by the different amplitudes of the 10.7 and 12-hour peaks in the non-tidal case compared to the tidal one, unlike the 6.7-hour peak which remains the same. The peaks at 12 hours and 10.7 hours are also visible in the model without tides, but with lower energy, thus confirming that these peaks are due to amplification of existing modes forced by winds and atmospheric pressure."

*Fig 6f. There's no sign of the 6.7 hour period here, which is surprising in comparison to figure 5b. It is in fig 6e too. I think this may be an artefact of the colour choice, because even in the tidal run the energy around 6 hours is much lower than at the longer periods. Is there is a better way to show this, for example normalising with frequency?*

We changed the colourbar on this figure to ensure the 6 hour peak is visible. As you said, it's there, but it's a few orders of magnitude lower than the tidal peaks.

*Figs 7, 8, 9 and in supplementary mat., since the depth axis is logarithmic, is it necessary to also have separate panels for depth ranges? Could you reduce clutter by combining them? Not essential, just a suggestion.*

We chose to separate the panels in order to demonstrate the differences between the layers of the Mediterranean (upper Atlantic water, intermediate Mediterranean water, and deep Mediterranean water), since we also refer to these layers in the text. Moreover, even with the logarithmic scale, it was easier to see the features this way than in a single panel.

*Section 6: I accept your previous reply to the reviewers about retaining section 6, if a little peripheral to the main paper.*

*Line 214, "Assessing [...] is evidence of" doesn't make sense, reword.*

We changed this sentence to the following: "Assessing the impact of tides on the mixed layer depth can provide indirect evidence for the impact of internal tides on the vertical mixing and vertical motion."

*Almost all figures: I have already flagged this, but many of the figures MUST be more legible before publication compared to the latest full manuscript. Please ensure numbers and text are large enough to read on every figure when the paper is printed on A4. (If you have good eyesight, I suggest checking it by printing at half size! Also check the detailed requirements in the instructions to authors.) The publication charges at this journal do not increase with bigger figures.*

We have updated all of the figures, increasing the font size in most of them and increasing their size to take up the full width of the page where appropriate. We also made the checks by printing the paper at half size as you suggested: everything is clearer now.

*All data should be available, please refer to the author instructions on data provision.*

We have made the data available on Zenodo (link: https://zenodo.org/records/10911630), and also submitted this link on the journal website along with the revised manuscript.

*Regards,*

*Dr Joanne Williams, editor.*